# Provably Secure PUF-Based Lightweight Mutual Authentication Scheme for Wireless Body Area Networks

SangCheol Lee [1], SuHwan Kim [1], SungJin Yu [2], NamSu Jho [2] and YoHan Park [1,*]

1 College of Engineering, Department of Computer Engineering, Keimyung University, Daegu 42601, Republic of Korea
2 Electronics and Telecommunications Research Institute, Daejeon 34129, Republic of Korea
* Correspondence: yhpark@kmu.ac.kr; Tel.: +82-53-580-5229

**Abstract:** Wireless body area networks (WBANs) are used in modern medical service environments for the convenience of patients and medical professionals. Owing to the recent COVID-19 pandemic and an aging society, WBANs are attracting attention. In a WBAN environment, the patient has a sensor node attached to him/her that collects patient status information, such as blood pressure, blood glucose, and pulse; this information is simultaneously transmitted to his/her respective medical professional through a gateway. The medical professional receives and checks the patient's status information and provides a diagnosis. However, sensitive information, including the patient's personal and status data, are transmitted via a public channel, causing security concerns. If an adversary intercepts this information, it could threaten the patient's well-being. Therefore, a secure authentication scheme is essential for WBAN environments. Recently, Chen et al. proposed a two-factor authentication scheme for WBANs. However, we found out Chen et al.'s scheme is vulnerable to a privileged insider, physical cloning, verification leakage, impersonation, and session key disclosure attacks. We also propose a secure physical-unclonable-function (PUF)-based lightweight mutual authentication scheme for WBANs. Through informal security analysis, we demonstrate that the proposed scheme using biometrics and the PUF is safe against various security attacks. In addition, we verify the security features of our scheme through formal security analyses using Burrows–Abadi–Needham (BAN) logic, the real-or-random (RoR) model, and the Automated Validation of Internet Security Protocols and Applications (AVISPA). Furthermore, we evaluate the security features, communication costs, and computational costs of our proposed scheme and compare them with those of other related schemes. Consequently, our scheme is more suitable for WBAN environments than the other related schemes.

**Keywords:** wireless body area networks; authentication; biometric; physical unclonable function; BAN logic; RoR model; AVISPA

## 1. Introduction

Recently, with the increasing number of elderly people in society, the demand for medical services is increasing, owing to the health problems of the aging society [1]. In addition, the emergence and spread of infectious diseases such as COVID-19 has accelerated this demand [2]. Therefore, solving the problem of meeting the supply and demand for healthcare has emerged as a challenge for governments in various countries. Many attempts have been made to use wireless sensor networks (WSNs) to address this problem. Because of sensor miniaturization and improved wireless communication technology, WSNs are widely used in various environments, such as the Industrial Internet of Things [3], smart homes [4], and healthcare [5]. A method was thus proposed that comprises a wireless body area network (WBAN) that incorporates WSNs into the medical field [6]. The WBAN framework includes medical professionals, gateways, and sensor nodes. Through a gateway, a medical professional receives information concerning a patient's condition from sensors attached

to the patient or elderly person's body [7]. Medical services that use WBANs are more efficient for both medical professionals and patients. Using them, medical professionals can conveniently treat more patients than before, and patients can receive treatment regardless of location. This approach also limited the spread of infectious diseases by reducing contact between medical professionals and patients during the COVID-19 pandemic. Therefore, research on WBANs has been conducted continuously.

In a WBAN, sensitive information, such as patient status and personal information, is transmitted to medical professionals using insecure channels. Thus, an adversary could steal information from these public channels and attempt security breaches, including replay, impersonation, and man-in-the-middle (MITM) attacks [8]. In addition, a medical professional's mobile device could be stolen, and an adversary could attempt to impersonate the rightful owner using the parameters extracted from the device through power analysis attacks. Furthermore, an adversary could physically capture the sensor node, extract the secret parameters, and impersonate it. If a malicious adversary succeeds in any of the aforementioned attacks and gains sensitive patient information, this may have a significant adverse effect on the patient, such as a misdiagnosis [9]. Therefore, the security of authentication schemes for WBANs is directly related to the well-being of the patient [10].

In 2021, Chen et al. [11] proposed a two-factor authentication scheme for related existing WBAN schemes. They asserted that their scheme, which uses a single hash, is lightweight, heterogeneous, and allows joint operations to prevent various security threats, such as sensor node capture, privileged insider, and stolen verifier attacks. However, we demonstrate that Chen et al.'s scheme cannot resist physical cloning, privileged insiders, verification table leakage, impersonation, and session key disclosure attacks. To overcome the security issues in Chen et al.'s scheme, we designed a secure physical-unclonable-function (PUF)-based three-factor mutual authentication scheme, which we use with a fuzzy extractor [12] to increase security.

### 1.1. Research Contributions

The contributions of this paper are as follows:

- We review Chen et al.'s scheme to demonstrate that it cannot prevent physical cloning, privileged insider, verification table leakage, impersonation, and session key disclosure attacks.
- We propose a secure PUF-based three-factor mutual authentication scheme to remedy the security vulnerabilities in Chen et al.'s scheme.
- We conducted an informal security analysis to demonstrate that our scheme is secure against various security hazards, including stolen/lost mobile devices, privileged insiders, physical cloning, and stolen verifier attacks.
- We analyzed the security features of the proposed scheme using the well-known Burrows–Abadi–Needham (BAN) logic and real-or-random (RoR) model, which improve the mutual authentication and session key security, respectively. Furthermore, we utilized the Automated Verification of Internet Security Protocols and Applications (AVISPA) simulation tool to prove that the proposed scheme is resistant to replay and man-in-the-middle attacks.
- We evaluated the communication costs, computational costs, and security features of our scheme. Consequently, our scheme provides lower communication and computational costs and higher security levels compared with the existing schemes.

### 1.2. Organization

In Section 2, we introduce related works for WMSNs. We describe the system model, adversary model, PUF, and fuzzy extractor in Section 3. We provide a review of Chen et al.'s scheme and cryptanalysis of their scheme in Sections 4 and 5. Then, we propose the secure authentication scheme on WBANs in Section 6. The security and performance analyses of our scheme are shown in Sections 7 and 8. Lastly, we present the paper's conclusion in Section 9.

## 2. Related Works

Various authentication schemes have been proposed for wireless medical sensor networks (WMSNs). Kumar et al. [13] (2012) presented an authentication scheme for healthcare applications using WMSNs. This scheme provides a secure session key establishment between users and medical sensor nodes and allows the users to change their passwords. However, in 2013, He et al. [14] demonstrated that Kumar et al.'s scheme could not withstand attacks such as offline password guessing and privileged insider attacks. In addition, they proved that Kumar et al.'s scheme did not guarantee anonymity. Accordingly, He et al. proposed a more secure scheme and asserted that their scheme is robust against various attacks. Unfortunately, in 2015, Wu et al. [15] demonstrated that He et al.'s scheme was vulnerable to offline password guessing, user impersonation, and sensor node capture attacks. Accordingly, they proposed an authentication scheme using a smart card to store sensitive information from medical professionals, which provides a higher level of security in the WMSN environment. In 2017, Li et al. [16] proposed an anonymous mutual authentication and key agreement scheme for WMSNs using hash operations and XOR operations, which was more efficient than previous related schemes. Unfortunately, in 2020, Gupta et al. [17] demonstrated that Li et al.'s scheme could not prevent intermediate node capture, sensor node impersonation, and hub node impersonation attacks. They also proved that Li et al.'s scheme was vulnerable to linkable sessions and traceability. Therefore, they proposed an authentication scheme in the WBAN environments that overcomes the security vulnerabilities of Li et al.'s scheme. In 2019, Ostad–Sharif et al. [18] proposed an authentication key agreement scheme consisting of three tiers for WBANs. Their scheme ensured anonymity to protect users' sensitive information. However, in 2020, Alzahrani et al. [19] claimed that Ostad et al.'s scheme is vulnerable to brute-force guessing attacks, and it is possible to compute all previous session keys. Subsequently, they presented an anonymous authenticated key exchange scheme with better security and efficiency to demonstrate the known weaknesses of Ostad et al.'s scheme.

Recently, PUF-based authentication schemes have been proposed for various environments to prevent attacks. In 2018, Mahalat et al. [20] proposed a PUF-based scheme that secures WiFi authentication for Internet of Things (IoT) devices and protects them against invasive, semi-invasive, or tampering attacks. In 2019, Zhu et al. [21] proposed a lightweight RFID mutual authentication scheme using a PUF. Their scheme provides secure authentication between the server and a tag. They asserted that their scheme could prevent clone attacks because a PUF cannot be duplicated. In 2021, Mahmood et al. [22] suggested a mutual authentication and key exchange scheme for multiserver-based device-to-device (D2D) communication. The entire process of Mahmood et al.'s scheme uses only XOR operations and hash functions, and PUF is introduced to protect against physical capture attacks. In the same year, Chuang et al. [23] proposed a PUF-based authenticated key exchange scheme for IoT environments. Their scheme did not require verifiers or explicit challenge–response pairs (CRPs). Therefore, IoT nodes can freely authenticate each other and generate a session key without the assistance of any verifier or server. Kwon et al. [24] proposed a three-factor-based mutual authentication and key agreement scheme with a PUF for WMSNs. They proved that their scheme could protect against physical cloning attacks using a PUF.

In 2020, Fotouhi et al. [25] proposed a two-factor authentication scheme for WBANs and asserted that it was safe against sensor node capture attacks. Unfortunately, in 2021, Chen et al. [11] demonstrated that the aforementioned scheme is vulnerable to sensor node attacks and proposed an improved security-enhanced two-factor authentication scheme for WBANs. However, we discovered that their scheme is insecure against privileged insider attacks, physical cloning attacks, verification table leakage attacks, etc. Therefore, we propose a secure PUF-based lightweight mutual authentication scheme for WBANs that resolves these security issues.

## 3. Preliminaries

This section introduces the general system model, the threat model, and relevant mathematical preliminaries including the PUF and fuzzy extractor, which can improve our scheme's security.

### 3.1. System Model

Figure 1 shows the general system model of a WBAN, which consists of medical professionals such as doctors and nurses, sensor nodes, and a gateway. The details are as follows:

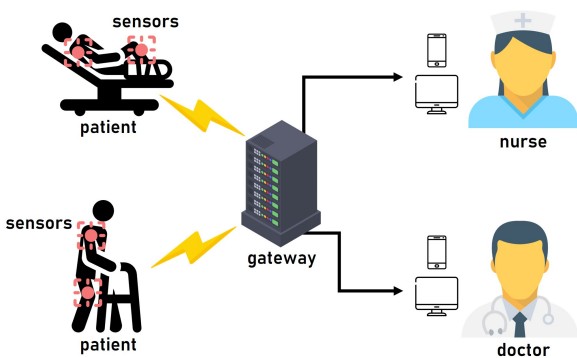

**Figure 1.** The general system model of WBANs.

- User ($U_i$): A user who wants to use the WBAN services receives a smart card from the gateway. After registration, the user can receive information from the sensor node attached to the patient's body.
- Gateway ($GW_j$): The gateway acts as a relay that connects patients with medical professionals. The gateway stores the value required for authentication.
- Sensor node ($SN_k$): The sensor node must be authenticated by the gateway. The authenticated sensor node is attached to the patient's body and transmits information to the medical professionals.

### 3.2. Adversary Model

To analyze the security of the proposed scheme, we applied the widely used Dolev–Yao (DY) adversary model. Under the DY model, a malicious adversary can inject, eavesdrop, modify, or delete messages transmitted using public channels. We also adopted the Canetti and Krawczyk (CK) adversary model to analyze the proposed scheme. The CK model is relatively strong compared with the DY model and is widely used to analyze scheme security. In the CK model, the adversary can intercept a random value and generate the master key of a gateway:

- An adversary can steal a medical professional's smart device and use a power analysis attack to extract sensitive information inside the cell phone.
- An adversary can obtain a patient's sensor node and extract important information within the sensor node through a physical cloning attack.
- An adversary can be a privileged insider, so it can also obtain a registration message from medical professionals
- An adversary can perform various attacks, such as password guessing, stolen verifier, and man-in-the-middle attacks.

### 3.3. Physical Unclonable Function

PUFs are physical circuits that operate using only a one-way function. The PUF circuit uses an input–output bit-string pair termed the "challenge–response pair". Even if numerous challenges are encountered in a PUF circuit, each has a unique output response.

In this paper. We express this process as $R = PUF(C)$, where $R$ and $C$ are a response and a challenge. The PUF's properties are as follows:

- The PUF is an unclonable circuit.
- The circuit of the PUF is easy to implement.
- The output of the PUF is unpredictable.
- The output of the PUF depends only on a physical circuit.

If the same challenge is entered into the PUF circuit of the same device, the same output response is printed. However, if a challenge is introduced into the PUF from different devices, different output responses are printed. Thus, the PUF provides a unique one-way function that cannot be replicated. The ability of the PUF to resist replication makes it impossible for adversaries to succeed with various attacks, such as physical cloning attacks.

### 3.4. Fuzzy Extractor

In this section, the purpose and basic concepts of the fuzzy extractor are discussed. However, biometric information is vulnerable to noise. Therefore, it is difficult to obtain a constant response value. Consequently, before users can utilize their biometrics, the biometric noise must be eliminated, for which we used a fuzzy extractor. The details are given below:

- $Gen(Bio_i) = <\sigma_i, \tau_i>$: This algorithm is intended to generate keys using biometric information. It receives biometric information as a parameter and returns the secret key data $R_i$ and a public reproduction $P_i$ as a helper value.
- $Rep(Bio_i^*, \tau_i) = \sigma_i$: This algorithm is for reproducing secret data $R_i$. The input of this algorithm is biometric information $Bio_i^*$ and $P_i$. The algorithm returns the secret key $R_i$ as a result.

## 4. Review of Chen et al.'s Scheme

In 2021, Chen et al. [11] proposed a two-factor authentication scheme for WBANs. Their scheme provides sensor node registration, user registration and mutual authentication, and a key exchange phase. The notations used in the Chen et al.s scheme are also presented in Table 1.

**Table 1.** Notations and definitions of Chen et al.'s scheme.

| Notation | Definition |
| --- | --- |
| $U_i$ | $i$-th user |
| $ID_i, PW_i$ | identity of $U_i$, password of $U_i$ |
| $GW_j$ | $j$-th gateway |
| $GID_j, G_j$ | identity of $GW_j$, secret key of $GW_j$ |
| $SN_k, SID_k$ | $k$-th sensor, its identity |
| $CID_i, QID_k$ | Temporary pseudoidentity of $U_i$ and $SN_k$ |
| $N_l$ | Network identifier of sensor set |
| $M_i$ | $i$-th message |
| $SG_k$ | Shared key between sensor and gateway |
| $SK_u$ | Session key generated by user |
| $SK_g$ | Session key generated by gateway |
| $SK_s$ | Session key generated by sensor node |
| $R_s, R_0, R_u, R_g, R_x, R_y, R_z$ | Temporary random number |
| $Gen(.)$ | Fuzzy biometric generator |
| $Rep(.)$ | Fuzzy biometric reproduction |
| $BIO_i$ | Biometric template of the user |
| $h(.)$ | Hash function |
| $\|\|$ | Concatenation operator |
| $\oplus$ | Exclusive-OR operator |

### 4.1. User Registration Phase

A medical professional such as a doctor or nurse must register in the gateway to use this network system. We describe the sensor node registration phase below:

**Step 1:** The user enters her/his own $ID_i$, $PW_i$ and imprints $Bio_i$ into the mobile device. Then, $U_i$ calculates $Gen(Bio_i) = \ <\sigma_i, \tau_i>$, $HPW_i = h(PW_i||\sigma_i)$ and sends $ID_i$, $HPW_i$ as a registration request to the gateway through a secure channel.

**Step 2:** Upon receiving $ID_i$, $PW_i$ determines whether the identity is new. If it is new, $GW_j$ calculates $CID_i = h(ID_i)$ and stores $CID_i$, $HPW_i$. Then, $GW_j$ selects a secret random number $R_0$. After that, $GW_j$ computes $A_1 = h(CID_i||GID_j||R_0 \oplus G_j) \oplus HPW_i$ and $A_2 = h(GID_j||HPW_i) \oplus (R_0 \oplus G_j)$ and stores $A_1$ in memory. Finally, $GW_j$ sends $\{A_2, GID_i\}$ to $U_i$ via a secure channel.

**Step 3:** $U_i$ computes $A_3 = h(ID_i||HPW_i)$. Then, $U_i$ stores $\{A_2, A_3, GID_j, Gen(.), Rep(.), \tau_i\}$.

### 4.2. Sensor Node Registration Phase

The sensor node must be registered with the gateway to transmit the health information of the patient. We show the sensor node registration phase of Chen et al.'s scheme as follows:

**Step 1:** $SN_k$ sends $SID_k$ and $N_l$ over a secure channel.

**Step 2:** $GW_j$ determines whether $SID_k$ is a new identity and generates a new pseudoidentity $QID_k$. $GW_j$ computes $SG_k = h(SID_k||G_j \oplus N_l)$ and stores $\{QID_k, N_l\}$ in the memory. Then, $GW_j$ sends $\{SG_k, QID_k\}$ to $SN_k$ via a secure channel.

**Step 3:** $SN_k$ computes $RSG_k = SG_k \oplus SID_k$ and saves $\{RSG_k, QID_k\}$ in the memory.

### 4.3. Login Phase

A medical professional must log in to the mobile device to use this network system. The detailed steps are illustrated in Figure 2:

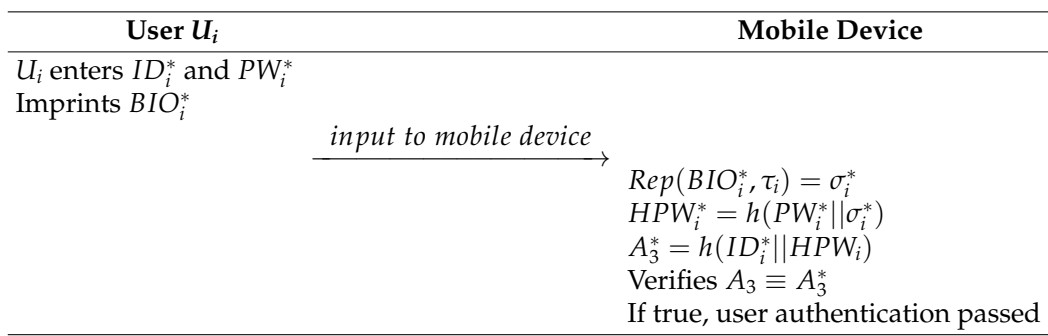

**Figure 2.** Login phase of Chen et al.'s scheme.

**Step 1:** $U_i$ enters his/her own $ID_i^*$, $PW_i^*$ and imprints $Bio_i^{'}$ into the mobile device.

**Step 2:** The mobile device computes $Rep(BIO_i^*, \tau_i) = \sigma_i^*$, $HPW_i^* = h(PW_i^*||\sigma_i^*)$, and $A_3^* = h(ID_i^*||HPW_i)$. Then, the mobile device verifies $A_3$ by comparison. If $A_3 = A_3^*$, the mobile device allows $U_i$ to log in.

### 4.4. Authentication and Key Agreement Phase

In this phase, the medical professionals and the sensor node conduct a mutual authentication and key agreement phase to authenticate each other and establish a session key. Figure 3 shows the authentication and key agreement phase of Chen et al.'s scheme, and the details are as follows:

| User $U_i$ | Gateway $GW_j$ | Sensor Node $SN_k$ |
|---|---|---|
| Selects $SID_k, R_u, T_1$ | | |
| Computes $(R_0 \oplus G_j) = A_2 \oplus h(GID_j||HPW_i)$ | | |
| $B_1 = SID_k \oplus h(GID_j||HPW_i)$ | | |
| $B_2 = R_u \oplus h(GID_j||HPW_i \oplus SID_k)$ | | |
| $B_3 = (R_0 \oplus G_j) \oplus h(GID_j||R_u)$ | | |

$$\xrightarrow{\quad M_1 = \{CID_i, GID_j, B_1, B_2, B_3, T_1\} \quad}$$

| | Verifies $|T_1 - T_c| \leq \Delta T$ | |
|---|---|---|
| | Gets $HPW_i, QID_k$ | |
| | Computes $SID_k = B_1 \oplus h(GID_j||HPW_i)$ | |
| | $R_u = B_2 \oplus h(GID_j||HPW_i \oplus SID_k)$ | |
| | $(R_0 \oplus G_j) = B_3 \oplus h(GID_j||R_u)$ | |
| | $A_1^* = h(CID_i||GID_j||R_0 \oplus G_j) \oplus HPW_i$ | |
| | Checks $A_1 \equiv A_1^*$ | |
| | Selects $R_g, T_2$ | |
| | $SG_k = h(SID_k||G_j \oplus N_l)$ | |
| | $B_4 = R_u \oplus HPW_i \oplus SG_k$ | |
| | $B_5 = R_g \oplus h(SG_k||SID_k)$ | |
| | $B_6 = h(QID_k||B_4||B_5||SG_k||R_u \oplus HPW_i||R_g)$ | |

$$\xrightarrow{\quad M_2 = \{QID_k, B_4, B_5, B_6, T_2\} \quad}$$

| | | Verifies $|T_2 - T_c| \leq \Delta T$ |
|---|---|---|
| | | Gets $RSG_k$ based on $QID_k$ |
| | | $SG_k = RSG_k \oplus SID_k$ |
| | | $(R_u \oplus HPW_i) = B_4 \oplus SG_k$ |
| | | $R_g = B_5 \oplus h(SG_k||SID_k)$ |
| | | $B_6^* = h(QID_k||B_4||B_5||SG_k||R_u \oplus HPW_i||R_g)$ |
| | | Verifies $B_6^* \equiv B_6$ |
| | | Selects $R_s, T_3$ |
| | | Computes $SK_s = h(R_u \oplus HPW_i||R_g||R_s)$ |
| | | $B_7 = h(SG_k||R_g) \oplus R_s$ |
| | | $B_8 = h(R_g||R_s||SG_k||T_3)$ |

$$\xleftarrow{\quad M_3 = \{B_7, B_8, T_3\} \quad}$$

| | Verifies $|T_3 - T_c| \leq \Delta T$ | |
|---|---|---|
| | Computes $R_s = h(SG_k||R_g) \oplus B_7$ | |
| | $B_8^* = h(R_g||R_s||SG_k||T_3)$ | |
| | Checks $B_8^* \equiv B_8$ | |
| | Selects $T_4$ | |
| | $SK_g = h(R_u \oplus HPW_i||R_g||R_s)$ | |
| | $B_9 = h(R_u \oplus GID_j||HPW_i) \oplus (R_g||R_s)$ | |
| | $B_{10} = h(R_0 \oplus G_j||SK_g||R_u)$ | |

$$\xleftarrow{\quad M_4 = \{B_9, B_{10}, T_4\} \quad}$$

| $|T_4 - T_c| \leq \Delta T$ | | |
|---|---|---|
| Computes $(R_g||R_s) = B_9 \oplus h(R_u \oplus GID_j||HPW_i)$ | | |
| $SK_u = h(R_u \oplus HPW_i||R_g||R_s)$ | | |
| $B_{10}^* = h(R_0 \oplus G_j||SK_u||R_u)$ | | |
| Checks $B_{10}^* \equiv B_{10}$ | | |
| If true, communication is possible | | |

**Figure 3.** Authentication and key agreement phase of Chen et al.'s scheme.

**Step 1:** $U_i$ selects the $SID_k$ of the sensor to be accessed, generates a random number $R_u$, and creates a timestamp $T_1$. Then, $U_i$ calculates $(R_0 \oplus G_j) = A_2 \oplus h(GID_j||HPW_i)$, $B_1 = SID_k \oplus h(GID_j||HPW_i)$, $B_2 = R_u \oplus h(GID_j||HPW_i \oplus SID_k)$, and $B_3 = (R_0 \oplus G_j) \oplus h(GID_j||R_u)$. Finally, $U_i$ sends message $M_1\{CID_i, GID_j, B_1, B_2, B_3, T_1\}$ to $GW_j$ via a public channel.

**Step 2:** $GW_j$ receives the message $M_1$ and verifies the legitimacy of $T_1$ by determining whether it matches $|T_1 - T_c| \leq \Delta T$. $GW_j$ retrieves the memory and obtains the $HPW_i, QID_k$ that matches $CID_i$ in $M_1$. $(SID_m||\alpha_m) = Dec_{MSK}(MID_m)$. Then, $GW_j$ computes $SID_k = B_1 \oplus h(GID_j||HPW_i)$, $R_u = B_2 \oplus h(GID_j||HPW_i \oplus SID_k)$, $(R_0 \oplus G_j) = B_3 \oplus h(GID_j||R_u)$, and $A_1^* = h(CID_i||GID_j||R_0 \oplus G_j) \oplus HPW_i$. $GW_j$ verifies $A_1 \equiv A_1^*$. If the verification is false, $GW_j$ stops the conversation. Otherwise, $GW_j$ confirms the justification of the identity of $U_i$, and it generates a random number $R_g$ and a new timestamp $T_2$. Then, $GW_j$ computes $SG_k = h(SID_k||G_j \oplus N_l)$, $B_4 = R_u \oplus HPW_i \oplus SG_k$, $B_5 = R_g \oplus h(SG_k||SID_k)$, and $B_6 = h(QID_k||B_4||B_5||SG_k||R_u \oplus HPW_i||R_g)$. Finally, $GW_j$ sends $M_2\{QID_k, B_4, B_5, B_6, T_2\}$ to $SN_k$ via a public channel.

**Step 3:** $SN_k$ receives the message $M_2$ and verifies that $|T_2 - T_c| \leq \Delta T$. The message is fresh if the verification is true. Then, $SN_k$ obtains the corresponding $RSG_k$ in storage based on $QID_k$. $SN_k$ computes $SG_k = RSG_k \oplus SID_k$, $(R_u \oplus HPW_i) = B_4 \oplus SG_k$, and $B_6^* = h(QID_k||B_4||B_5||SG_k||R_u \oplus HPW_i||R_g)$. Afterward, $GW_j$ verifies whether $B_6^* \equiv B_6$. If it is true, $SN_k$ generates a random number $R_s$ and a timestamp $T_3$. $SN_k$ calculates the keys $SK_s = h(R_u \oplus HPW_i||R_g||R_s)$, $B_7 = h(SG_k||R_g \oplus R_s)$, and $B_8 = h(R_g||R_s||SG_k||T_3)$. Then, $SN_k$ sends message $M_3\{B_7, B_8, T_3\}$ to $GW_j$ via a public channel.

**Step 4:** $GW_j$ receives the message $M_3$ and verifies the freshness of timestamp $T_3$ using $|T_3 - T_c| \leq \Delta T$. If the verification passes, $GW_j$ generates timestamp $T_4$ and calculates $R_s = h(SG_k||R_g) \oplus B_7$ and $B_8^* = h(R_g||R_s||SG_k||T_3)$, then verifies whether $B_8^* \equiv B_8$. If the verification is correct, $GW_j$ generates $T_4$ and calculates $SK_s = h(R_u \oplus HPW_i||R_g||R_s)$, $B_9 = h(R_u \oplus GID_j||HPW_i) \oplus (R_g||R_s)$, and $B_{10} = h(R_0 \oplus G_j||SK_g||R_u)$. After that, $GW_j$ sends message $M_4\{B_9, B_{10}, T_4\}$ to $U_i$ via a public channel.

**Step 5:** $U_i$ receives the message $M_4$ and verifies that $|T_2 - T_c| \leq \Delta T$. If the verification is true, the message is fresh. Then, $U_i$ computes $(R_g||R_s) = B_9 \oplus h(R_u \oplus GID_j||HPW_i)$, $SK_u = h(R_u \oplus HPW_i||R_g||R_s)$, and $B_{10}^* = h(R_0 \oplus G_j||SK_u||R_u)$. Finally, $U_i$ verifies whether $B_{10}^* \equiv B_{10}$, and if this is true, the verification and key exchange are a success.

## 5. Cryptanalysis of Chen et al.'s Scheme

In this section, we analyze the security defects of Chen et al.'s scheme. Our analysis shows that their scheme is vulnerable to privileged insider attacks, physical cloning attacks, and verification table leakage attacks. In addition, malicious adversary $\mathcal{A}$ can impersonate the user, sensor node, and gateway and disclose a session key.

### 5.1. Privileged Insider Attack

A privileged insider can support $\mathcal{A}$ by giving various important information such as registration message and values stored on the mobile device of the user. We describe the procedures are as follows:

**Step 1:** $\mathcal{A}$ can obtain a registration request message $\{ID_i, HPW_i\}$ and the secret parameter $\{A_2, A_3, GID_j, Gen(.), Rep(.), \tau_i\}$ extracted from the smart device of the user.

**Step 2:** The adversary $\mathcal{A}$ intercepts $M_1\{CID_i, GID_j, B_1, B_2, B_3, T_1\}$, and $M_3\{B_7, B_8, T_3\}$ transmitted by the public channel.

**Step 3:** $\mathcal{A}$ calculates $(R_0 \oplus G_j)^* = A_2 \oplus h(GID_j||HPW_i)$, $SID_k^* = B_1 \oplus h(GID_j||HPW_i)$, $R_u^* = B_2 \oplus h(GID_j||HPW_i \oplus SID_k)$, and $(R_g||R_s)^* = B_9 \oplus h(R_u \oplus GID_j||HPW_i)$. Then, $\mathcal{A}$ can extract the parameters $(R_0 \oplus G_j)^*$, $SID_k^*$, $R_u^*$, and $(R_g||R_s)^*$.

**Step 4:** $\mathcal{A}$ calculates $B_1^* = SID_k \oplus h(GID_j||HPW_i)$, $B_2^* = R_u \oplus h(GID_j||HPW_i \oplus SID_k)$, $B_3^* = (R_0 \oplus G_j) \oplus h(GID_j||R_u)$, and $SK_u = h(R_u \oplus HPW_i||R_g||R_s)$. Thereafter, $\mathcal{A}$ can generate $M_1\{CID_i, GID_j, B_1^*, B_2^*, B_3^*, T_1^*\}$ and send it to $GW_j$ by impersonating legitimate user $U_i$. In addition, $\mathcal{A}$ can calculate $SK_u^* = h(R_u \oplus HPW_i||(R_g||R_s)^*)$ to generate session key $SK_u^*$. Thus, $\mathcal{A}$ can disclose or exploit the session key.

Thus, Chen et al.'s scheme is insecure against privileged insider attacks.

### 5.2. Physical Cloning Attack

In this attack, we assume that $\mathcal{A}$ can clone sensor node $SN_k$ physically and extract the sensitive value $\{RSG_k, QID_k\}$ stored in the memory of $SN_k$. In order to be able to forward message $\{B_7, B_8, T_3\}$ on behalf of the legitimate $GW_j$ and generate session key $SK_s$, then $\mathcal{A}$ has to calculate the value of $B_7 = h(SG_k||R_g \oplus R_s)$, $B_8 = h(R_g||R_s||SG_k||T_3)$, and $SK_s = h(R_u \oplus HPW_i||R_g||R_s)$ through the following steps:

**Step 1:** The adversary $\mathcal{A}$ can obtain the messages $M_2\{QID_k, B_4, B_5, B_6, T_2\}$ and $M_3\{B_7, B_8, T_3\}$ by the eavesdropping attack.

**Step 2:** $\mathcal{A}$ computes $SG_k^*$ through $SG_k^* = RSG_k \oplus SID_k$.

**Step 3:** $\mathcal{A}$ calculates $(R_u \oplus HPW_i)^* = B_4 \oplus SG_k$, $R_s^* = B_5 \oplus h(SG_k || SID_k)$, and $R_s^* = h(SG_k || R_g) \oplus B_7$. Afterward, $\mathcal{A}$ obtains the parameters $(R_u \oplus HPW_i)^*$, $R_g^*$, and $R_s^*$.

**Step 4:** $\mathcal{A}$ can successfully compute $B_7^* = h(SG_k^* || R_g^*) \oplus R_s^*$, $B_8^* = h(R_g^* || R_s^* || SG_k^* || T_3^*)$, and $SK_s^* = h((R_u \oplus HPW_i)^* || R_g^* || R_s^*)$. Finally, $\mathcal{A}$ can generate authentication message $M_3^*\{B_7^*, B_8^*, T_3^*\}$ and session key $SK_s$.

Therefore, the scheme of Chen et al. cannot resist thephysical cloning attack.

### 5.3. Verification Table Leakage Attack

If $\mathcal{A}$ extracts the verification table $\{QID_k, N_l, CID_i, HPW_i, A_1\}$ of $GW_j$, $\mathcal{A}$ attempts to impersonate $GW_j$ and generate a session key. The details are described below:

**Step 1:** The malicious adversary $\mathcal{A}$ can obtain the messages $M_1\{CID_i, GID_j, B_1, B_2, B_3, T_1\}$, $M_2\{QID_k, B_4, B_5, B_6, T_2\}$, and $M_3\{B_7, B_8, T_3\}$ transmitted by the public channel.

**Step 2:** $\mathcal{A}$ computes $SID_k^* = B_1 \oplus h(GID_j || HPW_i)$, $R_u^* = B_2 \oplus h(GID_j || HPW_i \oplus SID_k^*)$, $(R_0 \oplus G_j)^* = B_3 \oplus h(GID_j || R_u^*)$, $SG_k^* = R_u^* \oplus HPW_i \oplus B_4$, $R_g^* = B_5 \oplus h(SG_k^* || SID_k^*)$, and $R_s^* = h(SG_k^* || R_g^*) \oplus B_7$ to generate parameters $SID_k^*, R_u^*, (R_0 \oplus G_j)^*, SG_k^*$, $R_g^*, R_s^*$.

**Step 3:** $\mathcal{A}$ calculates $B_4 = R_u \oplus HPW_i \oplus SG_k$, $B_5 = R_g \oplus h(SG_k || SID_k)$, $B_6 = h(QID_k || B_4 || B_5 || SG_k || R_u \oplus HPW_i || R_g)$, $SK_g^* = h(R_u^* \oplus HPW_i || R_g^* || R_s^*)$, $B_9^* = h(R_u^* \oplus GID_j || HPW_i) \oplus (R_g^* || R_s^*)$, and $B_{10}^* = h((R_0 \oplus G_j)^* || SK_g^* || R_u^*)$.

**Step 4:** Eventually, $\mathcal{A}$ can generate authentication messages $M_2^*\{QID_k, B_4^*, B_5^*, B_6^*, T_2^*\}$ and $M_4^*\{B_9^*, B_{10}^*, T_4^*\}$ and send them to the user and gateway disguised as a legal $GW_j$. Furthermore, $\mathcal{A}$ can generate session key $SK_g^*$ of $GW_j$ and adversely affect the system by exposing $SK_g^*$.

Therefore, Chen et al.'s scheme cannot withstand verification table leakage attacks.

### 5.4. Impersonation Attack

(1) User impersonation attack: In the previous privileged insider attack in Section 5.1, $\mathcal{A}$ can generate authentication message $M_1\{CID_i, GID_j, B_1^*, B_2^*, B_3^*, T_1^*\}$ and send it to the gateway to impersonate a legitimate user. Therefore, the scheme of Chen et al. is vulnerable to the user impersonation attack.

(2) Gateway impersonation attack: In the previous verification table attack in Section 5.3, $\mathcal{A}$ can calculate authentication messages $M_2^*\{QID_k, B_4^*, B_5^*, B_6^*, T_2^*\}$ and $M_4^*\{B_9^*, B_{10}^*, T_4^*\}$ and send them to the sensor node and user. However, the sensor node and gateway cannot recognize that the message transmitted from a gateway was not legal. Therefore, the scheme of Chen et al. cannot resist the gateway impersonation attack.

(3) Sensor node impersonation attack: In the previous physical cloning attack in Section 5.2, a malicious adversary $\mathcal{A}$ can compute message $M_3^*\{B_7^*, B_8^*, T_3^*\}$ to be sent to the gateway. However, the gateway recognizes that the message was transmitted from a legitimate sensor node. Therefore, Chen et al.'s scheme cannot withstand sensor node impersonation attacks.

### 5.5. Session Key Disclosure Attack

In the previous attacks, privileged insider in Section 5.1, physical cloning in Section 5.2, and verification table leakage in Section 5.3, $\mathcal{A}$ can generate session keys $SK_u, SK_k$, and $SK_g$. $\mathcal{A}$ attempts to exploit the generated session key to adversely affect the system and disclose it to the outside. Thus, the scheme of Chen et al. cannot prevent session key disclosure attacks.

## 6. Proposed Scheme

In this section, we propose a secure three-factor mutual authentication scheme for WBANs to overcome the security weaknesses of Chen et al.'s scheme. Our scheme also

considers the efficiency of the authentication process. Our scheme consists of user registration, sensor node registration, mutual authentication and key agreement, and password change phases. The notations and definitions used in the proposed scheme are explained in Table 2.

**Table 2.** Notations and definitionsof the proposed scheme.

| Notation | Definition |
|---|---|
| $U_i$ | $i$-th user |
| $ID_i, PW_i$ | identity of $U_i$, password of $U_i$ |
| $GW_j$ | $j$-th gateway |
| $GID_j, G_j$ | identity of $GW_j$, secret key of $GW_j$ |
| $SN_k, SID_k$ | $k$-th sensor, its identity |
| $CID_i$ | Temporary pseudoidentity of $U_i$ |
| $M_i$ | $i$-th message |
| $SG_k$ | Shared key between sensor and gateway |
| $SK_u$ | Session key generated by user |
| $SK_g$ | Session key generated by gateway |
| $SK_s$ | Session key generated by sensor node |
| $R_u, R_g, R_s, R_0, R_1, R_2$ | Temporary random number |
| $Gen(.)$ | Fuzzy biometric generator |
| $Rep(.)$ | Fuzzy biometric reproduction |
| $BIO_i$ | Biometric template of the user |
| $h(.)$ | Hash function |
| $\|\|$ | Concatenation operator |
| $\oplus$ | Exclusive-OR operator |

*6.1. User Registration Phase*

In order for a medical professional to receive patient information from the sensor node, he/she must be registered with the gateway in advance. The details are shown in Figure 4:

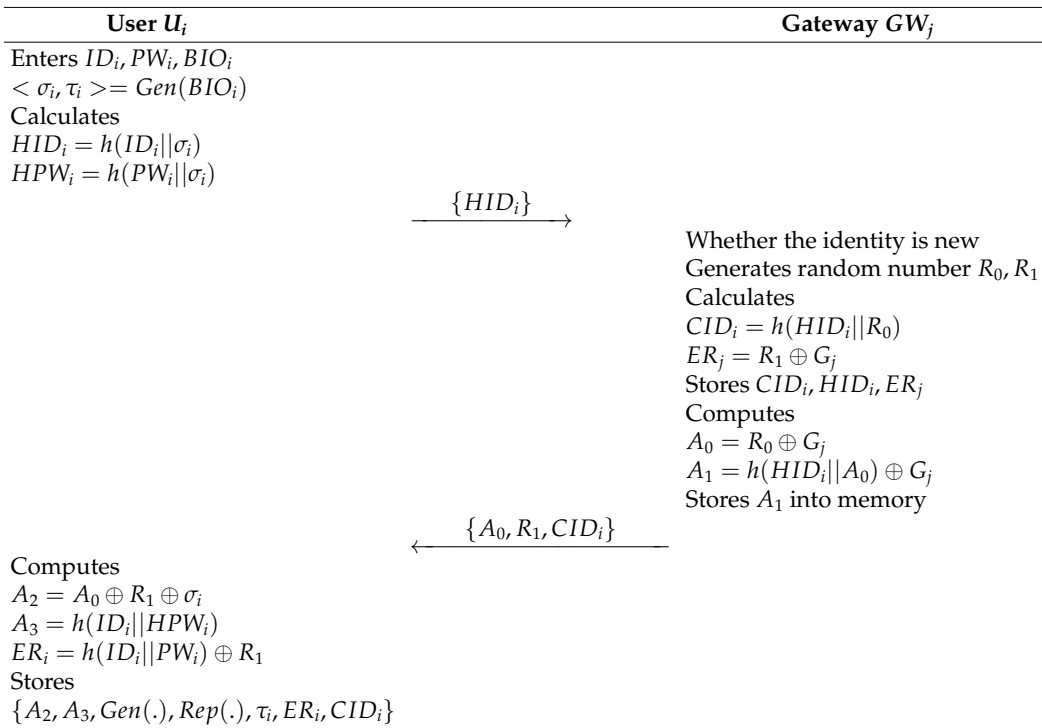

**Figure 4.** User Registration of the proposed scheme.

**Step 1:** $U_i$ inputs an identity $ID_i$, a password $PW_i$, and biometric template $BIO_i$ into the mobile device. Then, the mobile device computes $Gen(BIO_i) = < \sigma_i, \tau_i >$, $HID_i = h(ID_i||\sigma_i)$, and $HPW_i = h(PW_i||\sigma_i)$. $U_i$ sends $HID_i$ to the gateway through a secure channel.

**Step 2:** $GW_j$ receives $HID_i$ from $U_i$ and checks whether $HID_i$ is new. If it is new, $GW_j$ generates random numbers $R_0$ and $R_1$. Then, $GW_j$ calculates $CID_i = h(HID_i||R_0)$ and $ER_j = R_1 \oplus G_j$ and stores $CID_i, HID_i, ER_j$. Afterward, $GW_j$ computes $A_0 = R_0 \oplus G_j$ and $A_1 = h(HID_i||A_0) \oplus G_j$ and stores $A_1$ into memory. Finally, $GW_j$ sends message $\{A_0, R_1, CID_i\}$ to $U_i$ via a secure channel.

**Step 3:** $U_i$ receives message $A_0, R_1, CID_i$ from $GW_j$ and computes $A_2 = A_0 \oplus R_1 \oplus \sigma_i$, $A_3 = h(ID_i||HPW_i)$, and $ER_i = h(ID_i||PW_i) \oplus R_1$. Then, $GW_j$ stores $\{A_2, A_3, Gen(.), Rep(.), \tau_i, ER_i, CID_i\}$ in the mobile device.

*6.2. Sensor Node Registration Phase*

A sensor node must register with the gateway in order to transmit patient information to the medical professional. The sensor node registration phase is shown in Figure 5, and the detailed steps are as follows:

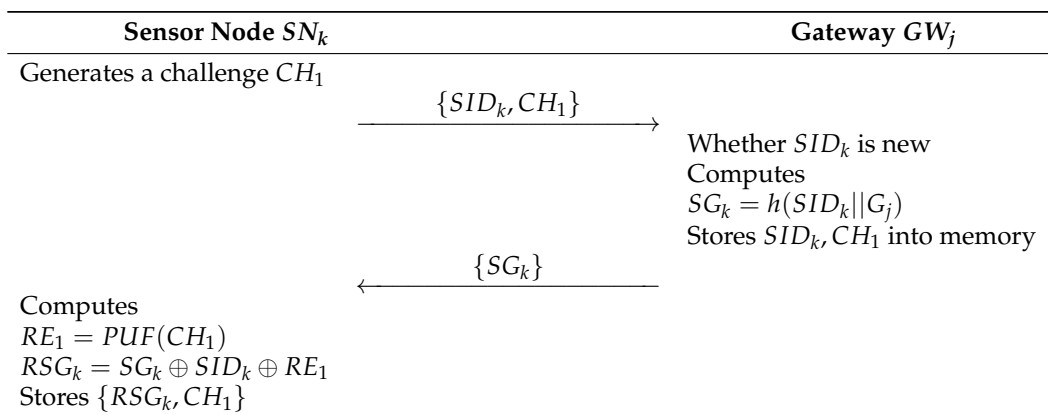

**Figure 5.** Sensor node registration of the proposed scheme.

**Step 1:** $SN_k$ generates a challenge $CH_1$ and sends identity $SID_k$ and $CH_1$ to $GW_j$ over a secure channel.

**Step 2:** $GW_j$ receives $SID_k$ and $CH_1$ from $SN_k$ and determines whether $SID_k$ is a new identity. If it is new, $GW_j$ computes $SG_k = h(SID_k||G_j)$ and stores $SID_k$ and $CH_1$ into memory. Then, $GW_j$ sends $SG_k$ to $SN_k$ through a secure channel.

**Step 3:** $SN_k$ receives $SG_k$ from $GW_j$. Then, $SN_k$ computes $RE_1 = PUF(CH_1)$ and $RSG_k = SG_k \oplus SID_k \oplus RE_1$ and saves $\{RSG_k, CH_1\}$ in the memory.

*6.3. Login Phase*

A medical professional must log in to the mobile device to utilize this WBAN system. The details are shown in Figure 6:

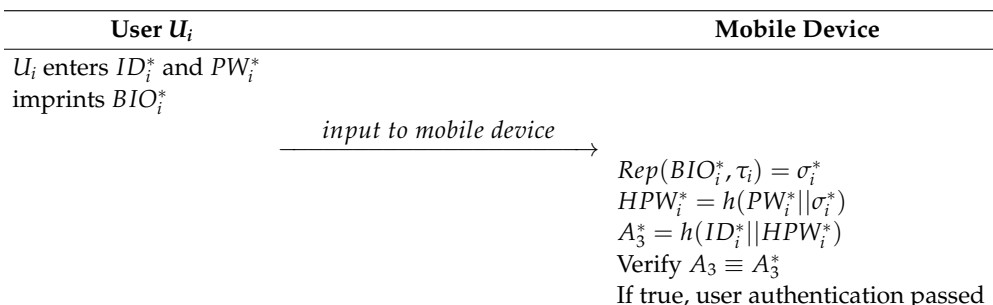

**Figure 6.** Login phase of the proposed scheme.

**Step 1:** $U_i$ enters $ID_i^*$ and $PW_i^*$ and imprints $BIO_i^*$ into the mobile device.

**Step 2:** The mobile device calculates $Rep(BIO_i^*, \tau_i) = \sigma_i^*$, $HPW_i^* = h(PW_i^* || \sigma_i^*)$, and $A_3^* = h(ID_i^* || HPW_i^*)$. Then, the mobile device verifies $A_3$ by comparison. If $A_3 = A_3^*$, $U_i$ logs in successfully.

*6.4. Mutual Authentication and Key Agreement Phase*

The medical professional sends an authentication message to the gateway and generates a session key among the medical professional, the sensor node, and the gateway. After that, the medical professionals can receive the patient's information from the sensor node. In Figure 7, we show the mutual authentication and key agreement phase of our scheme, and the details are given below:

**Step 1:** $U_i$ selects $SID_k, R_u, T_1$ and computes $R_1 = ER_i \oplus h(ID_i || PW_i)$ and $A_0 = A_2 \oplus R_1 \oplus \sigma_i$. Then, $U_i$ generates random nonce $R_u$ and calculates $B_1 = R_u \oplus R_1$, $B_2 = A_0 \oplus R_u \oplus R_1 \oplus HID_i$. Finally, $U_i$ sends $M_1\{SID_k, CID_i, B_1, B_2, T_1\}$ to $GW_j$ through a public channel.

**Step 2:** $GW_j$ receives message $M_1$ from $U_i$ and verifies that $|T_1 - T_c| \leq \Delta T$. If the verification passes, $GW_j$ checks whether $CID_i = CID_i^{old}$ or $CID_i = CID_i^{new}$. If ($CID_i == CID_i^{old}$), then it retrieves $\{HID_i^*, ER_j\}$ against $CID_i^{old}$, and if ($CID_i == CID_i^{new}$), it retrieves $\{HID_i^*, ER_j\}$ against $CID_i^{new}$. After that, $GW_j$ computes $R_1 = ER_j \oplus G_j$, $R_u = B_1 \oplus R_1$, $A_0 = B_2 \oplus R_u \oplus R_1 \oplus HID_i$, and $A_1^* = h(HID_i || A_0) \oplus G_j$. If $A_1 \overset{?}{=} A_1^*$ is true, $GW_j$ computes $CID_i^{new} = h(HID_i || R_u)$ and updates $CID_i^{new}$. Then, $GW_j$ selects $R_g, T_2$ and calculates $SG_k = h(SID_k || G_j)$, $C_1 = Ru \oplus HID_i$, $B_3 = C_1 \oplus SG_k \oplus CH_1$, $B_4 = R_g \oplus h(SG_k || SID_k)$, and $B_5 = h(B_4 || B_5 || SG_k || C_1 || R_g)$. Finally, $GW_j$ sends $M_2\{B_3, B_4, B_5, T_2\}$ to $SN_k$ via a public channel.

**Step 3:** $SN_k$ receives the message $M_2\{B_3, B_4, B_5, T_2\}$ and verifies the freshness of timestamp $T_2$ using $|T_2 - T_c| \leq \Delta T$. If the verification is true, the message is fresh. Then, $SN_k$ obtains the corresponding $RSG_k, CH_1$ and computes $RE_1 = PUF(CH_1)$, $SG_k = RSG_k \oplus SID_k \oplus RE_1$, $C_1 = B_3 \oplus SG_k \oplus CH_1$, $R_g = B_4 \oplus h(SG_k || SID_k)$, and $B_5^* = h(B_3 || B_4 || SG_k || C_1 || R_g)$. $SN_k$ verifies whether $B_5^* \overset{?}{=} B_5$. If verification is correct, $SN_k$ selects $R_s, T_3$ and computes $SK_s = h(C_1 || R_g || R_s)$, $B_6 = h(SG_k || R_g) \oplus R_s$, and $B_7 = h(R_g || R_s || SG_k || T_3 || C_1)$. $SN_k$ sends $M_3 = \{B_6, B_7, T_3\}$ to $GW_j$ through a public channel.

**Step 4:** $GW_j$ receives the message $M_3$ and verifies that $|T_3 - T_c| \leq \Delta T$. The message is fresh if the verification is true. Then, $GW_j$ computes $R_s = h(SG_k || R_g) \oplus B_6$ and $B_7^* = h(R_g || R_s || SG_k || T_3 || C_1)$. Afterward, $GW_j$ verifies whether $B_7^* \overset{?}{=} B_7$. If it is true, $GW_j$ selects $T_4$ and computes $SK_g = h(C_1 || R_g || R_s)$, $B_8 = R_u \oplus (R_g || R_s)$, and $B_9 = h(A_0 || SK_g || R_u)$. $GW_j$ sends $M_4 = \{B_8, B_9, T_4\}$ to $U_i$ via a public channel

**Step 5:** $U_i$ receives the message $M_4$ and verifies the legitimacy of $T_4$ by determining whether it matches $|T_4 - T_c| \leq \Delta T$. $U_i$ computes $(R_g || R_s) = B_8 \oplus R_u$, $SK_u = h(C_1 || R_g || R_s)$, and $B_9^* = h(A_0 || SK_u || R_u)$. Then, $U_i$ verifies whether $B_9^* \overset{?}{=} B_9$. If the verification is true, $U_i$ updates $CID_i^{new}$. Finally, the verification and key exchange are successful.

| User $U_i$ | Gateway $GW_j$ | Sensor Node $SN_k$ |
|---|---|---|

Selects $SID_k, R_u, T_1$
Computes
$R_1 = ER_i \oplus h(ID_i||PW_i)$
$A_0 = A_2 \oplus R_1 \oplus \sigma_i$
Generates random nonce $R_u$
$B_1 = R_u \oplus R_1$
$B_2 = A_0 \oplus R_u \oplus R_1 \oplus HID_i$

$$\xrightarrow{\quad M_1 = \{SID_k, CID_i, B_1, B_2, T_1\} \quad}$$

Verifies $|T_1 - T_c| \le \Delta T$
Checks whether
$CID_i = CID_i^{old}$ or $CID_i = CID_i^{new}$
if($CID_i == CID_i^{old}$)
{Retrieves $\{HID_i^*, ER_j\}$ against $CID_i^{old}$}
if($CID_i == CID_i^{new}$)
{Retrieves $\{HID_i^*, ER_j\}$ against $CID_i^{new}$}
Computes $R_1 = ER_j \oplus G_j$
$R_u = B_1 \oplus R_1$
$A_0 = B_2 \oplus R_u \oplus R_1 \oplus HID_i$
$A_1^* = h(HID_i||A_0) \oplus G_j$
Check $A_1 \stackrel{?}{=} A_1^*$
$CID_m^{new} = h(HID_i||R_u)$
Updates $CID_i^{new}$
Selects $R_g, T_2$
$SG_k = h(SID_k||G_j)$
$C_1 = R_u \oplus HID_i$
$B_3 = C_1 \oplus SG_k \oplus CH_1$
$B_4 = R_g \oplus h(SG_k||SID_k)$
$B_5 = h(B_4||B_5||SG_k||C_1||R_g)$

$$\xrightarrow{\quad M_2 = \{B_3, B_4, B_5, T_2\} \quad}$$

Verify $|T_2 - T_c| \le \Delta T$
Gets $RSG_k, CH_1$
$RE_1 = PUF(CH_1)$
$SG_k = RSG_k \oplus SID_k \oplus RE_1$
$C_1 = B_3 \oplus SG_k \oplus CH_1$
$R_g = B_4 \oplus h(SG_k||SID_k)$
$B_5^* = h(B_3||B_4||SG_k||C_1||R_g)$
Verify $B_5^* \stackrel{?}{=} B_5$
Selects $R_s, T_3$
Computes $SK_s = h(C_1||R_g||R_s)$
$B_6 = h(SG_k||R_g) \oplus R_s$
$B_7 = h(R_g||R_s||SG_k||T_3||C_1)$

$$\xleftarrow{\quad M_3 = \{B_6, B_7, T_3\} \quad}$$

Verify $|T_3 - T_c| \le \Delta T$
Computes $R_s = h(SG_k||R_g) \oplus B_6$
$B_7^* = h(R_g||R_s||SG_k||T_3||C_1)$
Check $B_7^* \stackrel{?}{=} B_7$
Selects $T_4$
$SK_g = h(C_1||R_g||R_s)$
$B_8 = R_u \oplus (R_g||R_s)$
$B_9 = h(A_0||SK_g||R_u)$

$$\xleftarrow{\quad M_4 = \{B_8, B_9, T_4\} \quad}$$

$|T_4 - T_c| \le \Delta T$
Computes $(R_g||R_s) = B_8 \oplus R_u$
$C_1 = R_u \oplus HID_i$
$SK_u = h(C_1||R_g||R_s)$
$B_9^* = h(A_0||SK_u||R_u)$
Checks $B_9^* \stackrel{?}{=} B_9$
Updates $CID_i^{new}$

**Figure 7.** Authentication and key agreement phase of the proposed scheme.

### 6.5. Password Update Phase

In our scheme, we provide an efficient password update process of the medical professional. We show the password update phase in Figure 8, and the detailed steps are as follows:

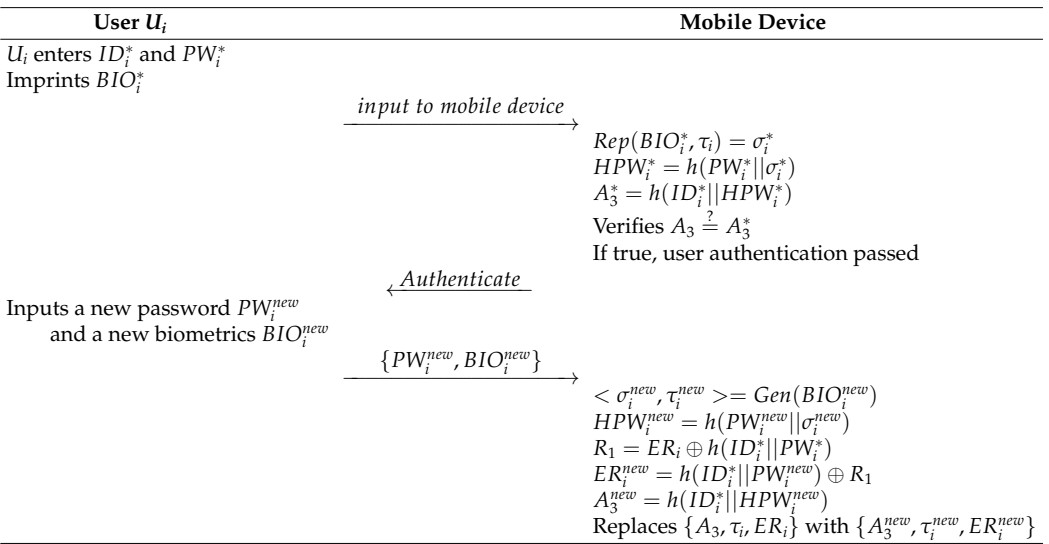

| User $U_i$ | Mobile Device |
|---|---|
| $U_i$ enters $ID_i^*$ and $PW_i^*$ | |
| Imprints $BIO_i^*$ | |
| | *input to mobile device* → |
| | $Rep(BIO_i^*, \tau_i) = \sigma_i^*$ |
| | $HPW_i^* = h(PW_i^*||\sigma_i^*)$ |
| | $A_3^* = h(ID_i^*||HPW_i^*)$ |
| | Verifies $A_3 \stackrel{?}{=} A_3^*$ |
| | If true, user authentication passed |
| | ← *Authenticate* |
| Inputs a new password $PW_i^{new}$ | |
| and a new biometrics $BIO_i^{new}$ | |
| | $\{PW_i^{new}, BIO_i^{new}\}$ → |
| | $<\sigma_i^{new}, \tau_i^{new}> = Gen(BIO_i^{new})$ |
| | $HPW_i^{new} = h(PW_i^{new}||\sigma_i^{new})$ |
| | $R_1 = ER_i \oplus h(ID_i^*||PW_i^*)$ |
| | $ER_i^{new} = h(ID_i^*||PW_i^{new}) \oplus R_1$ |
| | $A_3^{new} = h(ID_i^*||HPW_i^{new})$ |
| | Replaces $\{A_3, \tau_i, ER_i\}$ with $\{A_3^{new}, \tau_i^{new}, ER_i^{new}\}$ |

**Figure 8.** Password update phase of the proposed scheme.

**Step 1:** $U_i$ enters $ID_i^*$ and $PW_i^*$ and imprints $BIO_i^*$ to the mobile device.

**Step 2:** The mobile device calculates $Rep(BIO_i^*, \tau_i) = \sigma_i^*$, $HPW_i^* = h(PW_i^*||\sigma_i^*)$, and $A_3^* = h(ID_i^*||HPW_i^*)$ and verifies $A_3 \stackrel{?}{=} A_3^*$. If the equation is true, user authentication passes.

**Step 3:** $U_i$ inputs a new password $PW_i^{new}$ and a new biometric $BIO_i^{new}$ to the mobile device.

**Step 4:** The mobile device computes $Gen(BIO_i^{new}) = <\sigma_i^{new}, \tau_i^{new}>$, $HPW_i^{new} = h(PW_i^{new}||\sigma_i^{new})$, $R_1 = ER_i \oplus h(ID_i^*||PW_i^*)$, $ER_i^{new} = h(ID_i^*||PW_i^{new}) \oplus R_1$, and $A_3^{new} = h(ID_i^*||HPW_i^{new})$. Finally, the mobile device replaces $\{A_3, \tau_i, ER_i\}$ with $\{A_3^{new}, \tau_i^{new}, ER_i^{new}\}$.

## 7. Security Analysis

To prove the security features of the proposed scheme, we used BAN logic and the RoR model, which can prove the mutual authentication properties and session key security, respectively. Furthermore, we show that our scheme has resistance against man-in-the-middle and replay attacks using AVISPA. Furthermore, we claim that the proposed scheme can prevent various security attacks using informal analysis.

### 7.1. BAN Logic

In this section, BAN logic [26] is used to prove the mutual authentication of the proposed scheme. BAN logic uses a simple logic to explain the beliefs between the communication participants of authentication schemes. From that, many security schemes are proven by using BAN logic [27–29]. Table 3 shows the basic notation in BAN logic.

**Table 3.** Basic notations in BAN logic.

| Notation | Definition |
|----------|------------|
| $\mathcal{C}_1, \mathcal{C}_2$ | Principals |
| $\mathcal{T}_1, \mathcal{T}_2$ | Statements |
| $SK$ | Session key |
| $\mathcal{C}_1 \mid \equiv \mathcal{T}_1$ | $\mathcal{C}_1$ **believes** $\mathcal{T}_1$ |
| $\mathcal{C}_1 \mid \sim \mathcal{T}_1$ | $\mathcal{C}_1$ once **said** $\mathcal{T}_1$ |
| $\mathcal{C}_1 \Rightarrow \mathcal{T}_1$ | $\mathcal{C}_1$ **controls** $\mathcal{T}_1$ |
| $\mathcal{C}_1 \lhd \mathcal{T}_1$ | $\mathcal{C}_1$ **receives** $\mathcal{T}_1$ |
| $\#\mathcal{T}_1$ | $\mathcal{T}_1$ is **fresh** |
| $(\mathcal{T}_1)_K$ | $\mathcal{T}_1$ is **encrypted** with $K$ |
| $\mathcal{C}_1 \overset{K}{\leftrightarrow} \mathcal{C}_2$ | $\mathcal{C}_1$ and $\mathcal{C}_2$ have shared key $K$ |

7.1.1. Rules

We introduce five rules used in BAN logic:

1. Message meaning rule (MMR):

$$\frac{\mathcal{C}_1 \mid \equiv \mathcal{C}_1 \overset{K}{\leftrightarrow} \mathcal{C}_2, \quad \mathcal{C}_1 \lhd (\mathcal{T}_1)_K}{\mathcal{C}_1 \mid \equiv \mathcal{C}_2 \mid \sim \mathcal{T}_1};$$

2. Nonce verification rule (NVR):

$$\frac{\mathcal{C}_1 \mid \equiv \#(\mathcal{T}_1), \quad \mathcal{C}_1 \mid \equiv \mathcal{C}_2 \mid \sim \mathcal{T}_1}{\mathcal{C}_1 \mid \equiv \mathcal{C}_2 \mid \equiv \mathcal{T}_1};$$

3. Jurisdiction rule (JR):

$$\frac{\mathcal{C}_1 \mid \equiv \mathcal{C}_2 \Rightarrow \mathcal{T}_1, \quad \mathcal{C}_1 \mid \equiv \mathcal{C}_2 \mid \equiv \mathcal{T}_1}{\mathcal{C}_1 \mid \equiv \mathcal{T}_1};$$

4. Belief rule (BR):

$$\frac{\mathcal{C}_1 \mid \equiv (\mathcal{T}_1, \mathcal{T}_2)}{\mathcal{C}_1 \mid \equiv \mathcal{T}_1};$$

5. Freshness rule (FR):

$$\frac{\mathcal{C}_1 \mid \equiv \#(\mathcal{T}_1)}{\mathcal{C}_1 \mid \equiv \#(\mathcal{T}_1, \mathcal{T}_2)}.$$

7.1.2. Goals

The final goal of BAN logic in the proposed scheme is to achieve mutual authentication by agreeing on the session key $SK$. We define $U_i$, $GW_j$, and $SN_k$ as the user, gateway, and sensor node, respectively:

**Goal 1:** $U_i \mid \equiv GW_j \overset{SK}{\leftrightarrow} U_i$;

**Goal 2:** $U_i \mid \equiv GW_j \mid \equiv GW_j \overset{SK}{\leftrightarrow} U_i$;

**Goal 3:** $GW_j \mid \equiv GW_j \overset{SK}{\leftrightarrow} U_i$;

**Goal 4:** $GW_j \mid \equiv U_i \mid \equiv GW_j \overset{SK}{\leftrightarrow} U_i$;

**Goal 5:** $SN_k| \equiv GW_j \overset{SK}{\longleftrightarrow} SN_k;$

**Goal 6:** $SN_k| \equiv GW_j| \equiv GW_j \overset{SK}{\longleftrightarrow} SN_k;$

**Goal 7:** $GW_j| \equiv GW_j \overset{SK}{\longleftrightarrow} SN_k;$

**Goal 8:** $GW_j| \equiv SN_k| \equiv GW_j \overset{SK}{\longleftrightarrow} SN_k.$

### 7.1.3. Idealized Forms

In the proposed scheme, $M_1 = \{SID_k, CID_i, B_1, B_2, T_1\}$, $M_2 = \{B_3, B_4, B_5, T_2\}$, $M_3 = \{B_6, B_7, T_3\}$, and $M_4 = \{B_8, B_9, T_4\}$ are transmitted through public channels. We restructure the messages to fit the BAN logic, named "idealized forms":

$\mathcal{T}_1: U_i \rightarrow GW_j : \{R_u, A_0, HID_i, T_1\}_{R_1};$

$\mathcal{T}_2: GW_j \rightarrow SN_k : \{R_g, C_1, T_2\}_{SG_k};$

$\mathcal{T}_3: SN_k \rightarrow GW_j : \{R_s, T_3\}_{SG_k};$

$\mathcal{T}_4: GW_j \rightarrow U_i : \{R_g, R_s, T_4\}_{R_u}.$

### 7.1.4. Assumptions

The assumptions in the proposed scheme are shown as below:

$\mathcal{S}_1: GW_j| \equiv \#(T_1);$

$\mathcal{S}_2: SN_k| \equiv \#(T_2);$

$\mathcal{S}_3: GW_j| \equiv \#(T_3);$

$\mathcal{S}_4: U_i| \equiv \#(T_4);$

$\mathcal{S}_5: U_i| \equiv GW_j \Rightarrow (GW_j \overset{SK}{\longleftrightarrow} U_i);$

$\mathcal{S}_6: GW_j| \equiv U_i \Rightarrow (GW_j \overset{SK}{\longleftrightarrow} U_i);$

$\mathcal{S}_7: GW_j| \equiv SN_k \Rightarrow (GW_j \overset{SK}{\longleftrightarrow} SN_k);$

$\mathcal{S}_8: SN_k| \equiv GW_j \Rightarrow (GW_j \overset{SK}{\longleftrightarrow} SN_k);$

$\mathcal{S}_9: GW_j| \equiv GW_j \overset{R_1}{\longleftrightarrow} U_i;$

$\mathcal{S}_{10}: GW_j| \equiv GW_j \overset{SG_k}{\longleftrightarrow} SN_k;$

$\mathcal{S}_{11}: SN_k| \equiv GW_j \overset{SG_k}{\longleftrightarrow} SN_k;$

$\mathcal{S}_{12}: U_i| \equiv GW_j \overset{R_u}{\longleftrightarrow} U_i.$

### 7.1.5. BAN Logic Proof

**Step 1:** We can obtain $PR_1$ based on the first message $T_1$, and we obtain the following:

$$PR_1: GW_j \lhd \{R_u, A_0, HID_i, T_1\}_{R_1};$$

**Step 2:** Based on the message meaning rule, $PR_1$, and $\mathcal{S}_9$, we can obtain the following:

$$PR_2: GW_j| \equiv U_i| \sim (R_u, A_0, HID_i, T_1);$$

**Step 3:** Based on the freshness rule, $PR_2$, and $\mathcal{S}_1$, we can obtain the following:

$$PR_3: GW_j| \equiv \#(R_u, A_0, HID_i, T_1);$$

**Step 4:** Based on the nonce verification rule, $PR_2$, and $PR_3$, we obtain the following:

$$PR_4: GW_j| \equiv U_i| \equiv (R_u, A_0, HID_i, T_1);$$

**Step 5:** Based on the second message $T_2$, we obtain the following:

$$PR_5: SN_k \lhd \{R_g, C_1, T_2\}_{SG_k};$$

**Step 6:** Based on the message meaning rule, $PR_5$, and $\mathcal{S}_{11}$, we can obtain the following:

$$PR_6: SN_k| \equiv GW_j| \sim (R_g, C_1, T_2);$$

**Step 7:** Based on the freshness rule, $PR_6$, and $\mathcal{S}_2$, we can obtain the following:

$$PR_7: SN_k| \equiv \#(R_g, C_1, T_2);$$

**Step 8:** Based on the nonce verification rule, $PR_6$, and $PR_7$, we can obtain the following:

$$PR_8: SN_k| \equiv GW_j| \equiv (R_g, C_1, T_2);$$

**Step 9:** Based on the third message $T_3$, we can obtain the following:

$$PR_9: GW_j \lhd \{R_s, T_3\}_{SG_k};$$

**Step 10:** Based on the message meaning rule, $PR_9$, and $\mathcal{S}_{10}$, we can obtain the following:

$$PR_{10}: GW_j| \equiv SN_k| \sim (R_s, T_3);$$

**Step 11:** Based on the freshness rule, $PR_{10}$, and $\mathcal{S}_3$, we can obtain the following:

$$PR_{11}: GW_j| \equiv \#(R_s, T_3);$$

**Step 12:** Based on the nonce verification rule, $PR_{10}$, and $PR_{11}$, we can obtain the following:

$$PR_{12}: GW_j| \equiv SN_k| \equiv (R_s, T_3);$$

**Step 13:** Based on $PR_8$ and $PR_{12}$, $SN_k$ and $GW_j$ compute the session key $SK = h(C_1||R_g||R_s)$. Therefore, we can obtain the following goals:

$$PR_{13}: SN_k| \equiv GW_j| \equiv GW_j \overset{SK}{\longleftrightarrow} SN_k \quad \textbf{(Goal 6)}$$
$$PR_{14}: GW_j| \equiv SN_k| \equiv GW_j \overset{SK}{\longleftrightarrow} SN_k \quad \textbf{(Goal 8)};$$

**Step 14:** Based on the jurisdiction rule, $PR_{13}$, $PR_{14}$, $\mathcal{S}_7$, and $\mathcal{S}_8$, we can obtain the following goals:

$$PR_{15}: SN_k| \equiv GW_j \overset{SK}{\longleftrightarrow} SN_k \quad \textbf{(Goal 5)}$$
$$PR_{16}: GW_j| \equiv GW_j \overset{SK}{\longleftrightarrow} SN_k \quad \textbf{(Goal 7)};$$

**Step 15:** Based on the last message $T_4$, we can obtain the following:

$$PR_{17}: U_i \lhd \{R_g, R_s, T_4\}_{R_u};$$

**Step 16:** Based on the message meaning rule, $PR_{17}$, and $\mathcal{S}_{12}$, we can obtain the following:

$$PR_{18}: U_i \equiv SN_k| \sim (R_g, R_s, T_4);$$

**Step 17:** Based on the freshness rule, $PR_{18}$, and $\mathcal{S}_4$, we can obtain the following:

$$PR_{19}: U_i| \equiv \#(R_g, R_s, T_4);$$

**Step 18:** Based on the nonce verification rule, $PR_{19}$, and $PR_{17}$, we can obtain the following:

$$PR_{20}: U_i| \equiv GW_j| \equiv (R_g, R_s, T_4);$$

**Step 19:** Based on $PR_4$ and $PR_{20}$, $U_i$ and $GW_j$ compute the session key $SK$. Therefore, we can obtain the following goals:

$$PR_{21}: U_i| \equiv GW_j| \equiv GW_j \xleftrightarrow{SK} U_i \quad \textbf{(Goal 2)}$$
$$PR_{22}: GW_j| \equiv U_i| \equiv GW_j \xleftrightarrow{SK} U_i \quad \textbf{(Goal 4)};$$

**Step 20:** Based on the jurisdiction rule, $PR_{21}$, $PR_{22}$, $\mathcal{S}_5$, and $\mathcal{S}_6$, we can obtain the following goals:

$$PR_{23}: U_i| \equiv GW_j \xleftrightarrow{SK} U_i \quad \textbf{(Goal 1)}$$
$$PR_{24}: GW_j| \equiv GW_j \xleftrightarrow{SK} U_i \quad \textbf{(Goal 3)}.$$

*7.2. RoR Model*

To prove the security of the session key, we utilized a formal proof named the "real-or-random" (ROR) model [30]. Firstly, we define the participants, adversary, and queries. In the proposed scheme, there are three entities that perform the authentication phase to establish the session key. These entities are instantiated as participants and applied to the ROR model: $EP_{US}^i$, $EP_{GW}^j$, $EP_{SN}^k$. Note that $i$, $j$, and $k$ are the instances of the user, gateway, and sensor node, respectively. Next, we define the adversary of the ROR model. The adversary can fully control the whole network, including modifying, deleting, hijacking, and intercepting messages. Moreover, we introduce queries that are utilized to reveal the session key security of the scheme. The details are as follows:

- $Exe(EP_{US}^i, EP_{GW}^j, EP_{SN}^k)$: This is a passive attack, where the adversary obtain messages exchanged through public channels.
- $CorrD(EP_{US}^i)$: The $CorrD$ query is an active attack. The adversary obtains secret parameters that are stored in the smart card of $EP_{US}^i$ using power analysis attack.
- $Snd(EP)$: When the adversary uses the $Snd$ query, the adversary transfers messages to $EP_{US}^i$, $EP_{GW}^j$, and $EP_{SN}^k$. Moreover, the adversary receives return messages from the participants.
- $Test(EP)$: An unbiased coin $c$ is tossed, and the adversary obtains the result of this query. If the result value of $c$ is 0, the session key is not fresh. If the result value of $c$ is 1, we can demonstrate that the session key is fresh and secure. Otherwise, a null value ($\perp$) is obtained.

Security Proof

**Theorem 1.** *We define the adversary and possibility of breaking the session key security as $\mathcal{M}$ and $\mathcal{A}_M(BP)$, respectively. In the ROR model, $\mathcal{M}$ tries to guess $SK = h(C_1||R_g||R_s)$ in polynomial time. To do this, we give a definition of hash and puf as the range space of the hash function and PUF, respectively. Moreover, $q_{hash}$, $q_{puf}$, and $q_{snd}$ are the number of hash, puf, and Snd queries, respectively. We define $C'$ and $s'$ as Zipf's parameter [31], and the number of bits in the biometrics is BIO.*

$$\mathcal{A}_M(BP) \leq \frac{q_{hash}^2}{|hash|} + \frac{q_{puf}^2}{|puf|} + 2max\{C'q_{snd}^{s'}, \frac{q_{snd}}{2^{BIO}}\}$$

**Proof.** In the proposed scheme, the ROR security proof consists of five games $G_n$ ($0 \leq n \leq 4$). $\mathcal{M}$ tries to compute the session key $SK$ in each game $G_k$, and we define this winning possibility as $WN_{G_k}$. Our ROR security proof is performed according to the method of [32–34]:

$G_0$: $\mathcal{M}$ begins the real attack. Thus, $\mathcal{M}$ picks a random bit $c$. Therefore, we obtain Equation (1) as follows.

$$\mathcal{A}_M(BP) = |2\mathcal{M}[WN_{G_0}] - 1|. \tag{1}$$

$G_1$: As we mentioned before, $\mathcal{M}$ can obtain all of the messages in the proposed scheme using the query *Exe*. Thus, $M_1$, $M_2$, $M_3$, and $M_4$ can be intercepted and $\mathcal{M}$ executes the *Test* query as Equation (2). The session key $SK$ is composed of $C_1 = R_u \oplus HID_i$, $R_g$, and $R_s$. Thus, $\mathcal{M}$ must know all of the random nonces and the secret parameter of $US$. This means that $\mathcal{M}$ cannot calculate $SK$.

$$|\mathcal{M}[WN_{G_1}]| = |\mathcal{M}[WN_{G_0}]|. \tag{2}$$

$G_2$: In this game, the *hash* and *Snd* queries are utilized. However, we used the "cryptographic hash function", which can overcome the hash collision problem in the proposed scheme. Thus, $\mathcal{M}$ has no advantage using the *hash* and *Snd* queries. We show the following inequation (3) by applying the birthday paradox [35].

$$|\mathcal{M}[WN_{G_2}] - \mathcal{M}[WN_{G_1}]| \leq \frac{q_{hash}^2}{|hash|}. \tag{3}$$

$G_3$: In $G_3$, $\mathcal{M}$ attempts to break the session key security using the *puf* query. However, it is impossible to guess or compute the PUF function according to Section 3.3. Therefore, we obtain the following Equation (4).

$$|\mathcal{M}[WN_{G_3}] - \mathcal{M}[WN_{G_2}]| \leq \frac{q_{puf}^2}{|puf|}. \tag{4}$$

$G_4$: In the final game $G_4$, $\mathcal{M}$ utilizes the *CorrD* query and obtains secret parameters $\{A_2, A_3, Gen(.), Rep(.), \tau_i, ER_i, CID_i\}$ from the smart card. In the proposed scheme, all of the parameters are masked in the user's identity, password, and biometrics. To calculate $SK$ using the secret parameters, $\mathcal{M}$ must guess $U_i$, $PW_i$, and $BIO_i$ at the same time. Since guessing them in polynomial time is obviously impossible, $\mathcal{M}$ cannot derive $SK$. We apply Zipf's law and obtain the following Equation (5).

$$|\mathcal{M}[WN_{G_4}] - \mathcal{M}[WN_{G_2}]| \leq max\{C'q_{snd}^{s'}, \frac{q_{snd}}{2^{BIO}}\} \tag{5}$$

After that, $\mathcal{M}$ obtains the result bits $b$. Moreover, we can set up the following Equation (6).

$$\mathcal{M}[WN_{G_4}] = \frac{1}{2} \tag{6}$$

Using (1) and (2), Equation (7) can be calculated.

$$\frac{1}{2}\mathcal{A}_M(BP) = |\mathcal{M}[WN_{G_0}] - \frac{1}{2}| = |\mathcal{M}[WN_{G_1}] - \frac{1}{2}| \tag{7}$$

From (6) and (7), Equation (8) can be calculated.

$$\frac{1}{2}\mathcal{A}_M(BP) = |\mathcal{M}[WN_{G_1}] - \mathcal{M}[WN_{G_4}]| \tag{8}$$

Using the triangular inequality, we can obtain the following Equation (9).

$$\begin{aligned}
\frac{1}{2}\mathcal{A}_M(BP) &= |\mathcal{M}[WN_{G_1}] - \mathcal{M}[WN_{G_4}]| \\
&\leq |\mathcal{M}[WN_{G_1}] - \mathcal{M}[WN_{G_3}]| \\
&\quad + |\mathcal{M}[WN_{G_3}] - \mathcal{M}[WN_{G_4}]| \\
&\leq |\mathcal{M}[WN_{G_1}] - \mathcal{M}[WN_{G_2}]| \\
&\quad + |\mathcal{M}[WN_{G_2}] - \mathcal{M}[WN_{G_3}]| \\
&\quad + |\mathcal{M}[WN_{G_3}] - \mathcal{M}[WN_{G_4}]|
\end{aligned} \tag{9}$$

$$\leq \frac{q_{hash}^2}{2|hash|} + \frac{q_{puf}^2}{2|puf|} + max\{C'q_{snd}^{s'}, \frac{q_{snd}}{2^{BIO}}\} \tag{10}$$

We obtain the resulting inequation by multiplying (10) by two.

$$\mathcal{A}_M(BP) \leq \frac{q_{hash}^2}{|hash|} + \frac{q_{puf}^2}{|puf|} + 2max\{C'q_{snd}^{s'}, \frac{q_{snd}}{2^{BIO}}\}.$$

Thus, we prove the Theorem. □

### 7.3. AVISPA Simulation

In this section, we utilize the AVISPA simulation tool [36,37] to verify the resistance against the replay and man-in-the-middle attacks of the proposed scheme. The AVISPA simulation tool verifies the authentication scheme through a code called "High-Level scheme Specification Language (HLPSL)" on the Linux OS. Afterwards, the HLPSL code is converted to "Intermediate Format (IF)" to perform security verification on the four backends ("On-the-Fly Model Checker (OFMC)", "Three Automata based on Automatic Approximations for Analysis of Security Protocol (TA4SP)", "SAT-based Model Checker (SATMC)", and "Constraint Logic-based Attack Searcher (CL-AtSe)"). In this paper, we used the CL-AtSe and OFMC backends because these backends can support the XOR operator. Finally, the result window, i.e., "Output Format (OF)", is shown, and we can demonstrate that the proposed scheme can resist the replay and man-in-the-middle attacks if the OF summarizes the verification as "SAFE". We show the three basic roles of the proposed scheme: user $UI$, gateway $GWJ$, and sensor node $SNK$. The session, environment, and goals are shown in Figure 9. We also show the role of $UI$ in Figure 10.

```
role session(UI, GWJ, SNK : agent, SKuigw, SKsngw : symmetric_key, PUF,H : hash_func)

def=
local SN1, SN2, SN3, RV1, RV2, RV3 : channel(dy)
composition
user(UI, GWJ, SNK, SKuigw, SKsngw, PUF, H, SN1, RV1)
/\ gateway(UI, GWJ, SNK, SKuigw, SKsngw, PUF, H, SN2, RV2)
/\ sensornode(UI, GWJ, SNK, SKuigw, SKsngw, PUF, H, SN3, RV3)

end role

%%%%%%%%%%%%%%%%%%%%%%%%%%%%%%%%%%%%%%%

role environment()
def=
const ui, gwj, snk : agent,
            puf, h : hash_func,
            skuigw, sksngw : symmetric_key,
            ui_gw_ru, ui_sn_ru, gw_sn_rg, gw_ui_rg, sn_gw_rs, sn_ui_rs : protocol_id,
            sp1, sp2, sp3, sp4 : protocol_id,
            idi, cidi, sidk : text
intruder_knowledge = {h, idi, cidi, sidk}
composition
session(ui, gwj, snk, skuigw, sksngw, puf, h)
/\session(i, gwj, snk, skuigw, sksngw, puf, h)
/\session(ui, i, snk, skuigw, sksngw, puf, h)
/\session(ui, gwj, i, skuigw, sksngw, puf, h)

end role

%%%%%%%%%%%%%%%%%%%%%%%%%%%%%%%%%%%%%%%

goal
secrecy_of sp1, sp2, sp3, sp4
authentication_on ui_gw_ru
authentication_on ui_sn_ru
authentication_on gw_sn_rg
authentication_on gw_ui_rg
authentication_on sn_gw_rs
authentication_on sn_ui_rs

end goal

environment()
```

**Figure 9.** Role specification for the session, environment, and goals.

```
%%%AVISPA Simulation
role user(UI, GWJ, SNK : agent, SKuigw, SKsngw : symmetric_key, PUF,H : hash_func, SND,RCV : channel(dy))

played_by UI
def=
local State : nat,
              IDi, PWi, HIDi, HPWi, BIOi, R0, R1, CIDi, ERj, A0, A1, Gj, A2, A3, ERi : text,
              SIDk, CH1, SGk, RE1, RSGk, Ru, Rs, Rg, T1, T2, T3, T4, C1, B1, B2, B3, B4, B5, B6, B7, B8, B9, SK : text
              const sp1, sp2, sp3, sp4, ui_gw_ru, ui_sn_ru, gw_sn_rg, gw_ui_rg, sn_gw_rs, sn_ui_rs : protocol_id

init State := 0
transition
%%%%%%%%User registration phase
1. State = 0 /\ RCV(start) =|>
State' := 1
/\ HIDi' := H(IDi. BIOi)
/\ HPWi' := H(PWi. BIOi)
/\ SND({HIDi'}_SKuigw)
/\ secret({HIDi'}, sp1, {UI, GWJ})

2. State = 1 /\ RCV({A0. R1. H(H(IDi. BIOi). R0)}_SKuigw) =|>
State' := 2
/\ A2' := xor(xor(A0, R1), BIOi)
/\ A3' := H(IDi. H(PWi. BIOi))
/\ ERi' := xor(H(IDi. PWi), R1)
%login and authentication phase
/\ Ru' := new()
/\ T1' := new()
/\ B1' := xor(Ru', R1)
/\ B2' := xor(xor(xor(A0, Ru'), R1), H(IDi. BIOi))
/\ SND(SIDk. H(H(IDi. BIOi). R0). B1'. B2'. T1')
/\ witness(UI,GWJ,ui_gw_ru,Ru')
/\ witness(UI,SNK,ui_sn_ru,Ru')

3. State = 2 /\ RCV(xor(xor(Ru', Rs'), Rg'). H(A0. H(xor(Ru'. H(IDi. BIOi)). Rg'. Rs'). Ru'). T4') =|>
State' := 3
/\SK' := H(xor(Ru'. H(IDi. BIOi)). Rg'. Rs')
/\request(GWJ,UI,gw_ui_rg,Rg')
/\request(SNK,UI,sn_ui_rs,Rs')

end role
```

**Figure 10.** Role specification for the user.

In State 1, $UI$ receives the start message and computes $HID_i$ and $HPW_i$. Then, $UI$ sends $\{HID_i\}$ to $GWJ$. $GWJ$ registers $UI$ and returns $\{A_0, R_1, CID_i\}$ through a secure channel. State 2 is the login and authentication phase, for which $UI$ generates $R_u$, $T_1$ and computes the authentication request message $\{SID_k, CID_i, B_1, B_2, T_1\}$ to $GWJ$. At the same time, $UI$ generates function $witness(UI, GWJ, ui_gw_ru, Ru')$ and $witness(UI, SNK, ui_sn_ru, Ru')$, which means the proof of random nonce $R_u$'s freshness. Finally, $UI$ receives $\{B_8, B_9, T_4\}$ and computes the session key $SK = h(C_1||R_g||R_s)$. We verified the proposed scheme in the CL-AtSe and OFMC backends, and the result window is shown in Figure 11. Therefore, the proposed scheme can resist the replay and man-in-the-middle attacks.

```
% OFMC                                      SUMMARY
% Version of 2006/02/13                       SAFE
SUMMARY
  SAFE                                      DETAILS
DETAILS                                       BOUNDED_NUMBER_OF_SESSIONS
  BOUNDED_NUMBER_OF_SESSIONS                  TYPED_MODEL
PROTOCOL
  /home/span/span/testsuite/results/SANG.if  PROTOCOL
GOAL                                          /home/span/span/testsuite/results/SANG.if
  as_specified
BACKEND                                      GOAL
  OFMC                                         As Specified
COMMENTS
STATISTICS                                   BACKEND
  parseTime: 0.00s                             CL-AtSe
  searchTime: 6.31s
  visitedNodes: 1480 nodes                   STATISTICS
  depth: 12 plies
                                               Analysed   : 0 states
                                               Reachable  : 0 states
                                               Translation: 0.10 seconds
                                               Computation: 0.00 seconds
```

**Figure 11.** The AVISPA simulation result of the proposed scheme.

### 7.4. Informal Analysis

In this section, we demonstrate the security features of our proposed scheme, including those that resist against privileged insider, insider, physical, cloning, verification table leakage, impersonation, session key disclosure, ephemeral secret leakage, replay, man-in-the-middle, stolen mobile device, offline password guessing, and denial-of-service attacks. Moreover, the proposed scheme can provide user anonymity and perfect forward secrecy.

#### 7.4.1. User Anonymity

In our scheme, $\mathcal{A}$ cannot obtain the legitimate $U_i's$ identity $ID_i$, and even $\mathcal{A}$ extracts values $\{A_2, A_3, Gen(.), Rep(.), \tau_i, ER_i, CID_i\}$ inside $U_i's$ mobile device. $ID_i$ is masked by a hash function with $U_i's$ biometric information or $PW_i$ such that $HID_i = h(ID_i||\sigma_i)$, $A_3 = h(ID_i||HPW_i)$, and $ER_i = h(ID_i||PW_i) \oplus R_1$.

#### 7.4.2. Privileged Insider Attack

We can assume privileged insider $\mathcal{A}$ obtains the registration request message $\{HID_i\}$ of the medical professional. Furthermore, $\mathcal{A}$ can extract the parameters $\{A_2, A_3, Gen(.), Rep(.), \tau_i, ER_i, CID_i\}$ from the stolen mobile device of the medical professional using power analysis attack. $\mathcal{A}$ can also intercept transmitted messages such as $M_1$ and $M4$ on a public channel. After that, $\mathcal{A}$ attempts to impersonate a medical professional. To calculate authentication message $M_1\{SID_k, CID_i, B_1, B_2, T_1\}$, $\mathcal{A}$ must compute parameters $R_1$ and $A_0$. However, $\mathcal{A}$ cannot compute $R_1 = ER_i \oplus h(ID_i||PW_i)$ and $A_0 = A_2 \oplus R_1 \oplus \sigma_i$ because $\mathcal{A}$ cannot generate the $ID_i, PW_i$ and biometric information $BIO_i$ of $U_i$. Therefore, it is difficult for $\mathcal{A}$ to calculate the authentication message $M_1$ to impersonate a medical professional. $\mathcal{A}$ can also attempt to compute $SK_u = h(C_1||R_g||R_s)$. However, $\mathcal{A}$ cannot generate a session key of $U_i SK_u$. $\mathcal{A}$ cannot calculate $(R_g||R_s) = B_8 \oplus R_u$ and $R_u = B1 \oplus R_1$. In conclusion, the proposed scheme can resist the privileged insider attack.

#### 7.4.3. Insider Attack

Suppose that $U_i$ registers with $GW_j$ as a legal user and intercepts the transmitted messages such as $M_2$, $M_3$, and $M_4$. However, $U_i$ cannot calculate important parameters such as the symmetric key $SG_k$ shared by $GW_j$ and $SN_k$. Thus, $U_i$ cannot attempt various attacks, including the impersonate and session key disclosure attacks. As as result, our scheme can prevent the insider attack.

#### 7.4.4. Physical Cloning Attack

Assume that an adversary $\mathcal{A}$ physically captures a sensor node $SN_k$ and attempts to authenticate with $GW_j$ by disguising it as $SN_k$. $\mathcal{A}$ physically clones $SN_k$ to obtain a values $\{RSG_k, CH_1\}$ in the memory of $SN_k$ and intercepts authentication request messages $M_2$ on the public channel. Then, $\mathcal{A}$ attempts to generate authenticate message $M_3\{B_6, B_7, T_3\}$. However, $\mathcal{A}$ cannot generate a message $M_3$ because he/she cannot calculate the parameter $RE_1$ necessary to generate message $M_3$. $\mathcal{A}$ can replicate the same $CH_1$ from $SN_k$, but cannot generate the same $RE_1$. The PUF circuit cannot be forged. Thus, our scheme can withstand the physical cloning attack.

#### 7.4.5. Verification Table Leakage Attack

Suppose that $\mathcal{A}$ intercepts $\{CID_i, HID_i, ER_j, A_1, SID_k, CH_1\}$ in $GW_j's$ verification table of $GW_j$. Then, $\mathcal{A}$ eavesdrops the transmitted messages such as $M_1, M_2, M_3$ and intercepts message $M_4$ via an insecure channel. After that, $\mathcal{A}$ attempts to compute authentication request messages $M_2$ or $SK_g = h(C_1||R_g||R_s)$. However, $\mathcal{A}$ cannot calculate $SG_k = h(SID_k||G_j)$, which is essential for generating $M_2$ and $SK_g$, because $GW_j's$ secret key $G_j$ is unknown. Therefore, $\mathcal{A}$ cannot generate both $M_2$ and $SK_G$. As a result, our scheme can protect against verification table leakage attack.

### 7.4.6. Impersonation Attack

(1) User impersonation attack: For this attack, suppose an adversary $\mathcal{A}$ attempts to impersonate $U_i$. $\mathcal{A}$ must generate a valid authentication request message $M_1\{SID_k, CID_i, B_1, B_2, T_1\}$. $\mathcal{A}$ can extract $CID_i$ from $U_i's$ mobile device and intercept message $M_1\{SID_k, CID_i, B_1, B_2, T_1\}$ through a public channel, but cannot calculate the remaining values $\{B_1, B_2\}$ because $U_i's$ $ID_i$, $PW_i$, and $BIO_i$ are essential for calculating the remaining values $\{B_1, B_2\}$. Therefore, the proposed scheme is resilient against the user impersonation attack.

(2) Gateway impersonation attack: Suppose malicious adversary $\mathcal{A}$ tries to impersonate $GW_j$ and sends a authentication request message $M_2\{B_3, B_4, B_5, T_2\}$ to $SN_k$. To do this, $\mathcal{A}$ eavesdrops the transmitted messages $M_1$ and $M_2$. However, without having credentials $SG_k, C_1, HID_i, CH_1$, it is an impossible task for $\mathcal{A}$ to compute $M_2\{B_3, B_4, B_5, T_2\}$. Hence, the proposed scheme provides protection against the gateway impersonation attack.

(3) Sensor node impersonation attack: A malicious adversary $\mathcal{A}$ can try to impersonate $SN_k$. To do this, $\mathcal{A}$ intercepts transmitted messages $M_2$ and $M_3$ via an insecure channel and calculates the key agreement message $M_3\{B_6, B_7, T_3\}$. However, since $PUF(.)$ is a physically unclonable circuit, $\mathcal{A}$ cannot calculate $RE_1 = PUF(CH_1)$ and $SG_k = RSG_k \oplus SID_k \oplus RE_1$. Therefore, $\mathcal{A}$ cannot generate message $M_3\{B_6, B_7, T_3\}$. Thus, the proposed scheme prevents the sensor node impersonation attacks.

### 7.4.7. Session Key Disclosure Attack

If $\mathcal{A}$ tries to calculate a legitimate session key $SK = h(C_1||R_g||R_s)$, the adversary must obtain $HID_i, R_u, R_g, R_s$. However, $\mathcal{A}$ cannot obtain these values. $R_u, R_g,$ and $R_s$ are temporary random nonces used in a session, and $HID_i$ is masked as the legitimate $U_i's$ biometric information $BIO_i$. Hence, the proposed scheme provides protection against the session key disclosure attacks.

### 7.4.8. Perfect Forward Secrecy

$\mathcal{A}$ obtains long-term secret keys $\{SG_k, G_j\}$ and intercepts transmitted message $\{M_1, M_2, M_3, M_4\}$ through a public channel. After that, $\mathcal{A}$ attempts to generate $M_4$ to impersonate $GW_j$ or calculate $SK_g = h(C_1||R_g||R_s)$ to exploit the session key. However, $\mathcal{A}$ cannot compute the parameters $C_1$ without $U_i's$ identity $HID_i$ and random nonce $R_u$. For these reasons, our scheme provides perfect forward secrecy.

### 7.4.9. Ephemeral Secret Leakage Attack

$\mathcal{A}$ obtains random numbers $\{R_u, R_g, R_s, R_0, R_1, R_2\}$. After that, $\mathcal{A}$ attempts to compute the session key $SK_G = h(C_1||R_g||R_s)$. Unfortunately, $\mathcal{A}$ cannot generate session key $SK$ because $\mathcal{A}$ cannot calculate $C_1 = R_u \oplus HID_i$, which is essential for generating a session key $SK$. Thus, the proposed scheme can prevent the ESL attacks.

### 7.4.10. Replay and Man-in-the-Middle Attack

We assume that $\mathcal{A}$ eavesdrop transmitted message $\{M_1, M_2, M_3, M_4\}$ through a public channel. However, $\mathcal{A}$ cannot impersonate $U_i$, $GW_j$, and $SN_k$ by sending a message again. Because timestamps and random numbers such as $\{T_1, T_2, T_3, R_u, R_g, R_s\}$ are essential to generate a message, and the transmitted message is verified by $\{T_1, T_2, T_3, R_u, R_g, R_s\}$. Therefore, our scheme can prevent replay and man-in-the-middle attack.

### 7.4.11. Stolen Mobile Device Attack

Suppose that $\mathcal{A}$ succeeds in extracting stored values $\{A_2, A_3, Gen(.), Rep(.), \tau_i, ER_i, CID_i\}$ from $U_i's$ stolen mobile device. However, $\mathcal{A}$ cannot compute any meaningful value from $U_i$. The values stored in the mobile device are masked with $ID_i$, $PW_i$, and $BIO_i$ such as $A_2 = A_0 \oplus R_1 \oplus \sigma_i$, $A_3 = h(ID_i||HPW_i)$, $ER_i = h(ID_i||PW_i) \oplus R_1$. Therefore, $\mathcal{A}$ cannot attempt any attack. Thus, our scheme can resist the stolen mobile device attacks.

### 7.4.12. Offline Password Guessing Attack

$\mathcal{A}$ obtains $U_i's$ mobile device and extracts parameters $\{A_2, A_3, Gen(.), Rep(.), \tau_i, ER_i,$ $CID_i\}$ using the power analysis attack. After that, $\mathcal{A}$ tries to guess the password of $U_i$ using the extracted parameters. However, $\mathcal{A}$ cannot guess the $U_i's$ password $PW_i$ because the password is masked by the $U_i's$ $ID_i$, $BIO_i$, or random nonce $R_1$ such as $HPW_i = h(PW_i||\sigma_i)$, $A_3 = h(ID_i||HPW_i)$, and $ER_i = h(ID_i||PW_i) \oplus R_1$. Therefore, the proposed scheme is secure against the offline password guessing attacks.

### 7.4.13. Denial-of-Service

Assume that malicious $\mathcal{A}$ attempts to send $M_1\{SID_k, CID_i, B_1, B_2, T_1\}$ to $GW_j$ as a replay message. To do this, $\mathcal{A}$ must verify the value of $A_3 = h(ID_i||HPW_i)$ and pass the login phase. However, $\mathcal{A}$ cannot calculate a valid $A_3$ because $\mathcal{A}$ cannot obtain $ID_i$ and $HPW_i$. Therefore, $\mathcal{A}$ cannot transmit a replay message $M_1$ to $GW_j$. Thus, the proposed scheme is secure against the denial-of-service attacks.

### 7.4.14. Untraceability

Suppose a malicious $\mathcal{A}$ obtains $U_i's$ pseudoidentity $CID_i$. However, $\mathcal{A}$ cannot attempt any attack with the obtained $CID_i$. Every session, $GW_j$ updates the $CID_i$ stored with a $CID_i^{new}$ using random nonce $R_u$ after verifying that it is a legitimate user through $A_1 \stackrel{?}{=} A_1^*$ verification. For this reason, the proposed scheme ensures untraceability.

### 7.4.15. Mutual Authentication

To ensure mutual authentication, our scheme verifies that each entity is justified by $A_1 \stackrel{?}{=} A_1^*$, $B_5 \stackrel{?}{=} B_5^*$, $B_7 \stackrel{?}{=} B_7^*$, and $B_9 \stackrel{?}{=} B_9^*$. Moreover, all entities have verified freshness of messages through random values $R_u$, $R_g$, and $R_s$ generated by each entity. When the verification processes are passed, the entities are authenticated with each other. Therefore, our scheme achieves mutual authentication.

## 8. Performance

In this section, we evaluate the security features, communication costs, and computational costs of our scheme compared with the related schemes [11,38–41].

### 8.1. Security Features Comparison

We compared the performance of the proposed scheme with the related existing schemes [11,38–41]. As shown in Table 4, we considered various security functionalities and attacks, including "user anonymity", "privileged-insider attack", "offline password guessing attack", "stolen mobile device attack", "denial-of-service attack", "replay attack", "man-in-the-middle attack", "mutual authentication", "session key security", "known session specific temporary information attack", "untraceability property", "server-independent password update phase", "physical cloning attack", "perfect forward secrecy", "impersonation attack", "session-specific random number leakage attack", and "stolen verifier attack". Therefore, our scheme offers functional features and security in comparison with the related schemes [11,38–41].

### 8.2. Communication Cost Comparison

In this section, we demonstrate the comparison analysis for the communication cost of the proposed scheme with related existing schemes [11,38–41]. According to [42], we define that the bit lengths for the SHA-256 hash output, random number, identity, password, PUF challenge–response, timestamp, and ECC point are 256, 256, 128, 128, 128, 32, and 320 bits, respectively. Therefore, the communication costs of the proposed scheme can be described as below:

**Table 4.** Security and functionality features' comparison with related schemes.

| Security Properties | [38] | [39] | [40] | [41] | [11] | Proposed |
|:---:|:---:|:---:|:---:|:---:|:---:|:---:|
| $SP1$ | × | ✓ | ✓ | × | × | ✓ |
| $SP2$ | × | ✓ | × | × | × | ✓ |
| $SP3$ | ✓ | ✓ | ✓ | × | ✓ | ✓ |
| $SP4$ | ✓ | ✓ | ✓ | × | ✓ | ✓ |
| $SP5$ | ✓ | ✓ | ✓ | ✓ | ✓ | ✓ |
| $SP6$ | × | ✓ | ✓ | ✓ | × | ✓ |
| $SP7$ | × | ✓ | ✓ | ✓ | × | ✓ |
| $SP8$ | ✓ | ✓ | ✓ | ✓ | ✓ | ✓ |
| $SP9$ | ✓ | × | × | ✓ | ✓ | ✓ |
| $SP10$ | ✓ | ✓ | ✓ | ✓ | ✓ | ✓ |
| $SP11$ | ✓ | ✓ | × | ✓ | ✓ | ✓ |
| $SP12$ | ✓ | ✓ | × | × | × | ✓ |
| $SP13$ | × | × | ✓ | × | × | ✓ |
| $SP14$ | × | ✓ | ✓ | ✓ | ✓ | ✓ |
| $SP15$ | × | ✓ | ✓ | × | × | ✓ |
| $SP16$ | × | ✓ | ✓ | ✓ | ✓ | ✓ |
| $SP17$ | ✓ | ✓ | ✓ | × | × | ✓ |

Note: $SP1$: user anonymity; $SP2$: privileged insider attack; $SP3$: offline password guessing attack; $SP4$: stolen mobile device attack; $SP5$: denial-of-service attack; $SP6$: replay attack; $SP7$: man-in-the-middle attack; $SP8$: mutual authentication; $SP9$: session key security; $SP10$: known session specific temporary information attack; $SP11$: untraceability property; $SP12$: server-independent password update phase; $SP13$: physical cloning attack; $SP14$: perfect forward secrecy; $SP15$: impersonation attack; $SP16$: session-specific random number leakage attack; $SP17$: stolen verifier attack; ✓: provides or supports the security/functionality feature. ×: does not provide or support the security/functionality feature.

- Message 1: The message $M_1 = \{SID_k, CID_i, B_1, B_2, T_1\}$ needs (128 + 256 + 256 + 256 + 32) = 928 bits;
- Message 2: The message $M_2 = \{B_3, B_4, B_5, T_2\}$ requires (256 + 256 + 256 + 32) = 800 bits;
- Message 3: The message $M_3 = \{B_6, B_7, T_3\}$ requires (256 + 256 + 32) = 544 bits;
- Message 4: The message $M_4 = \{B_8, B_9, T_4\}$ needs (256 + 256 + 32) = 544 bits.

Therefore, the total communication cost of our scheme is 928 + 800 + 544 + 544 = 2816 bits. We show the total communication cost of our scheme and other related scheme [11,38–41] in Table 5. As a result, Figure 12 illustrates that our scheme has more efficient communication costs than other related schemes.

**Table 5.** Comparison of communication costs required for AKA.

| Schemes | Communication Costs | Messages |
|:---|:---|:---|
| Li et al. [38] | 3584 bits | 4 messages |
| Shin et al. [39] | 4480 bits | 4 messages |
| Rangwani et al. [40] | 2816 bits | 4 messages |
| Masud et al. [41] | 3200 bits | 4 messages |
| Chen et al. [11] | 3072 bits | 4 messages |
| Proposed | 2816 bits | 4 messages |

*8.3. Computational Cost Comparison*

We evaluated the computational costs of our scheme. According to [24], we determined the comparative analysis for the computational cost of the proposed scheme with [11,38–41] in the AKA phase. According to [24], we define $T_H$, $T_{RNG}$, $T_{EM}$, $T_{EA}$, $T_F$, and $T_{PUF}$ as the hash function ($\approx 0.00023$ ms), random number generation ($\approx 0.0539$ ms), ECC multiplication ($\approx 0.2226$ ms), ECC addition ($\approx 0.00288$ ms), fuzzy extractor ($\approx 0.268$ ms), and PUF

operation time ($\approx$0.012 ms), respectively. Additional, we did not consider the execution time of Exclusive-OR ($\oplus$) operations because it is computationally negligible. Table 6 shows the detail.

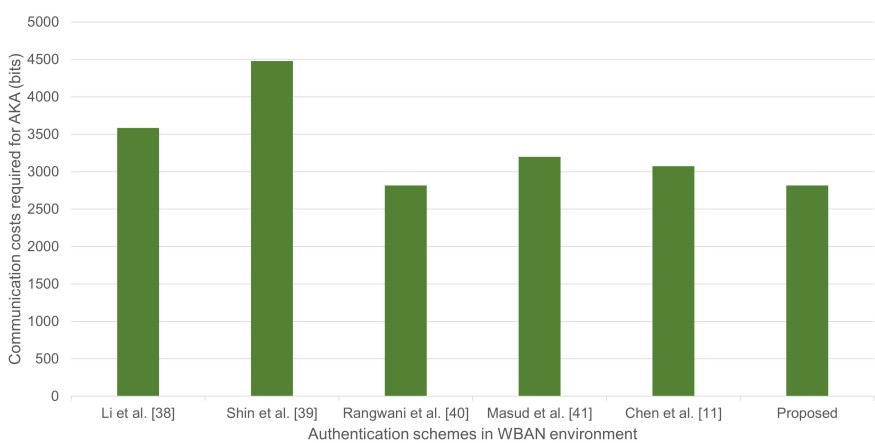

**Figure 12.** Communication cost comparison of related schemes [11,38–41].

The total computational costs of our scheme was estimated to be lower than other related schemes, except Masud et al.'s scheme. However, our scheme uses the fuzzy extractor and PUF to outperform Masud et al.'s scheme. Figure 13 shows that the computational cost (delay) increases with increasing numbers of users.

**Table 6.** Computational costs of each related scheme.

| Scheme | User | Gateway | Sensor Node | Total | Total Cost (s) |
|---|---|---|---|---|---|
| Li et al. [38] | $1T_{RNG} + 9T_H + 3T_{EM}$ | $1T_{RNG} + 8T_H + 1T_{EM}$ | $1T_{RNG} + 4T_H + 2T_{EM}$ | $3T_{RNG} + 21T_H + 6T_{EM}$ | $\approx$1.5021 ms |
| Shin et al. [39] | $1T_{RNG} + 1T_F + 14T_H + 2T_{EM}$ | $12T_H + 1T_{EM}$ | $1T_{RNG} + 5T_H + 1T_{EM}$ | $2T_{RNG} + 1T_F + 31T_H + 4T_{EM}$ | $\approx$1.232 ms |
| Rangwani et al. [40] | $5T_H + 2T_{EM} + 3T_{EA}$ | $4T_H + 2T_{EM} + 3T_{EA}$ | $8T_H + 2T_{EM} + 4T_{EA}$ | $17T_H + 6T_{EM} + 10T_{EA}$ | $\approx$1.36831 ms |
| Masud et al. [41] | $1T_{RNG} + 3T_H$ | $4T_{RNG} + 3T_H$ | $2T_{RNG} + 2T_H$ | $7T_{RNG} + 8T_H$ | $\approx$0.379 ms |
| Chen et al. [11] | $9T_H$ | $7T_H + 2T_{ENC}$ | $7T_H$ | $23T_H + 2T_{ENC}$ | $\approx$0.739 ms |
| Proposed | $5T_H + 1T_{RNG} + 1T_F$ | $9T_H + 1T_{RNG}$ | $5T_H + 1T_{RNG} + 1T_{PUF}$ | $19T_H + 3T_{RNG} + 1T_F + 1T_{PUF}$ | $\approx$0.44607 ms |

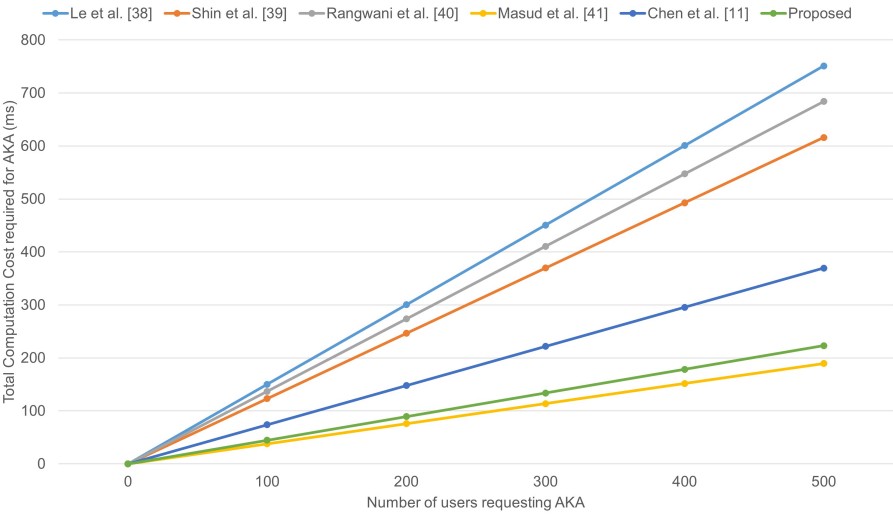

**Figure 13.** Total computation cost with increasing the AKA requests [11,38–41].

## 9. Conclusions

In this paper, we reviewed Chen et al.'s scheme and demonstrated that it is vulnerable to several attacks, such as privileged insider attacks, physical cloning attacks, verification leakage attacks, impersonation attacks, and session key disclosure attacks. Therefore, it is hard for Chen et al.'s scheme to be applied to WBANs properly, and a secure user authentication scheme should be presented for wireless medical environments. To enhance the security level of Chen et al.'s scheme, we proposed a secure three-factor mutual authentication and key agreement scheme using a secure PUF in the WBAN environment. Our scheme is lightweight because it uses only hash functions and Exclusive-OR operators and a fuzzy extractor to provide a secure login process. Moreover, our scheme resists physical cloning attacks using the PUF. The proposed scheme guarantees mutual authentication through BAN logic and utilizes the RoR model by which the session key is secured. Using the AVISPA simulation tool, we also demonstrated that our proposed scheme could withstand the replay and man-in-the-middle attacks. Moreover, we performed an informal security analysis to show that our proposed scheme provides protection against diverse hazards and attacks, including privileged insiders, physical cloning, verification table leakage, impersonation, session key disclosure, ephemeral secret leakage, replay, man-in-the-middle, stolen mobile device, offline password guessing, and denial-of-service attacks. We also proved that our scheme provides user anonymity, mutual authentication, and perfect forward secrecy. Finally, we compared the communication and computational costs of our scheme with those of related schemes after estimation. Based on the results, our scheme provides a lower communication cost and a higher security level compared to related existing schemes. Accordingly, we expect that our proposed scheme is to provide secure medical environments and to increase the use of the various healthcare applications.

**Author Contributions:** Conceptualization, S.L.; Formal analysis, S.Y. and Y.P.; Methodology, S.L. and S.K.; Software, S.Y.; Validation, N.J. and Y.P.; Formal Proof, Y.P.; Writing—original draft, S.L. and Y.P.; Writing—review and editing, S.K. and Y.P.; Supervision, N.J. and Y.P. All authors have read and agreed to the published version of the manuscript.

**Funding:** This work was supported by the Korean Government under Electronics and Telecommunications Research Institute (ETRI) Grant (20ZR1300, Core Technology Research on Trust Data Connectome).

**Institutional Review Board Statement:** Not applicable.

**Informed Consent Statement:** Not applicable.

**Data Availability Statement:** Not applicable.

**Conflicts of Interest:** The authors declare no conflict of interest.

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
