# Peer review of "Provably Secure PUF-Based Lightweight Mutual Authentication Scheme for Wireless Body Area Networks"

_electronics, doi:10.3390/electronics11233868_

Round 1

Reviewer 1 Report

Dear Authors,

Thank you very much for submitting your manuscript to this Journal for publication. As per editorial guidelines, this manuscript has been reviewed with intensive care. This paper has written well and it useful for the readers (i.e., academia and industrial engineers/ practitioners). However, there are several issues need to clarify before proceed to publications. These are listed as follows:

1) The present form of the abstract is a bit week and not much clear, I recommend to re-write it with 2/3 stronger sentences about your objectives/ findings that will give a better understanding for the readers. 

2) All the Fig. and Table captions sentences are not ended. It looks like a continuous sentence. It is recommended, to put a full stop (.) at the end of each sentence for all the Figure and Table captions. All the Figure captions are very short, it is better to write in detail caption. It seems that all the x- and y- axis need to write correctly for all Figures.

3) The current form of the conclusion is a bit week; I recommend to re-write the conclusion with 2/3 additional stronger sentences that you have achieved from this research. 

4) The references are 43 but it is recommended to add some suitable references from recent published works from this published and others too. So, need proper referencing. 

5) There are several typos and grammatical issues in the manuscript. Please clarify it carefully before proceed to publications.

6) Finally, I recommend to demonstrate in brief about the impact or significance of your research in the industry and community. 

Author Response

We thank you for your interest in our work and for constructive comments that will greatly improve the manuscript and we have tried to do our best to respond to the points raised.

Responses for your comments are attached.

Reviewer 2 Report

authors should include a greater number of references dated 2021 and 2022.

authors need to numerat ALL equations.

all equations must be cited in the text.

Author Response

(The authors gave the same response as above.)

Reviewer 3 Report

Thanks Authors for submitting the manuscript. The manuscript is very well organized. 

My only comment is from practical point of view. You know that there are many different technologies that can be used as the communication means between nodes, either professional devices or sensor nodes to the gateway. Each communication technology (e.g. WiFi, BLE, Zigbee, LoRa, NB-IoT and ...) has its own levels of securities. Although security procedures in different technologies might act in different network layers, hence may/(may not) overlap with your algorithm. Have you ever investigated the overlap of your algorithm with security procedures in specific communication technology?

If YES, Please mention that.

IF NO, it would be useful to somehow explain what is the relation between your algorithm and security procedures in communication technologies. I mean you may just want to mention overlaps, if there is any. 

Best Regards 

Author Response

(The authors gave the same response as above.)
